# Can Large Multimodal Models Understand Agricultural Scenes? Benchmarking with AgroMind

**Qingmei Li**[1*]**, Yang Zhang**[2,*]**, Zurong Mai**[2,*]**, Yuhang Chen**[2]**, Shuohong Lou**[2]**,**
**Henglian Huang**[2]**, Jiarui Zhang**[2]**, Zhiwei Zhang**[2]**, Yibin Wen**[2]**, Weijia Li**[2]**,**
**Haohuan Fu**[1,5]**, Jianxi Huang**[3,4]**, Juepeng Zheng**[2,5,†]

[1]Tsinghua Shenzhen International Graduate School    [2]Sun Yat-Sen University
[3]China Agricultural University    [4]Southwest Jiaotong University
[5]National Supercomputing Center in Shenzhen

## Abstract

Large Multimodal Models (LMMs) has demonstrated capabilities across various domains, but comprehensive benchmarks for agricultural remote sensing (RS) remain scarce. Existing benchmarks designed for agricultural RS scenarios exhibit notable limitations, primarily in terms of insufficient scene diversity in the dataset and oversimplified task design. To bridge this gap, we introduce **AgroMind**, a comprehensive agricultural remote sensing benchmark covering four task dimensions: spatial perception, object understanding, scene understanding, and scene reasoning, with a total of 13 task types, ranging from crop identification and health monitoring to environmental analysis. We curate a high-quality evaluation set by integrating nine public datasets and one private global parcel dataset, containing 28,482 QA pairs and 20,850 images. The pipeline begins with multi-source data pre-processing, including collection, format standardization, and annotation refinement. We then generate a diverse set of agriculturally relevant questions through the systematic definition of tasks. Finally, we employ LMMs for inference, generating responses, and performing detailed examinations. We evaluated 20 open-source LMMs and 4 closed-source models on AgroMind. Experiments reveal significant performance gaps, particularly in spatial reasoning and fine-grained recognition, it is notable that human performance lags behind several leading LMMs. By establishing a standardized evaluation framework for agricultural RS, AgroMind reveals the limitations of LMMs in domain knowledge and highlights critical challenges for future work. Data and code can be accessed at `https://rssysu.github.io/AgroMind/`.

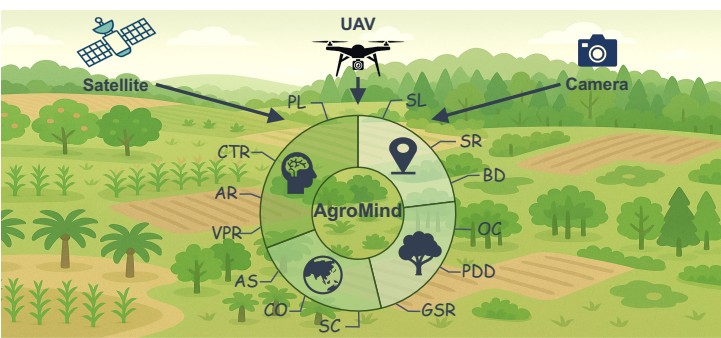

Figure 1: Overview of the AgroMind conceptual framework for benchmarking agricultural scenarios.

*These authors contributed equally to this work. †Corresponding: zhengjp8@mail.sysu.edu.cn

39th Conference on Neural Information Processing Systems (NeurIPS 2025) Track on Datasets and Benchmarks.

# 1 Introduction

Agriculture, as the cornerstone of global food security and economic stability, sustains the livelihoods of the global population and is essential to achieve the Sustainable Development Goal 2 (Zero Hunger) [1, 2]. However, it confronts unprecedented challenges from climate extremes and land degradation. Remote sensing (RS), with its multi-source data integration and real-time monitoring capabilities, provides critical support for crop growth monitoring [3–5], pest and disease detection [6, 7], and land use change evaluation [8, 9], driving the transformation of agriculture toward intelligence.

Agricultural RS typically involves heterogeneous multi-source data (e.g. optical, SAR, and LiDAR). Traditional deep learning methods struggle with cross-modal fusion, few-shot generalization, and dynamic environmental adaptation [10]. While recent advancements in large multimodal models (LMMs) have significantly enhanced visual understanding and reasoning [11–14]. As the requirements for visual processing in real-world applications continue to rise, numerous LMMs have been employed to improve the understanding of remote sensing images [15, 16]. The establishment of systematic benchmarking frameworks plays a pivotal role in comprehensively evaluating the core capabilities and latent potential of LMMs.

Despite the impressive capabilities demonstrated by LMMs across various domains, comprehensive benchmarks for agricultural RS remain scarce. Existing benchmarks in the RS field primarily focus on urban scenarios [17–19], and those designed for agricultural scenarios exhibit notable limitations, primarily in terms of **insufficient scene diversity** in the dataset and **oversimplified task design**, as shown in Fig. 2.

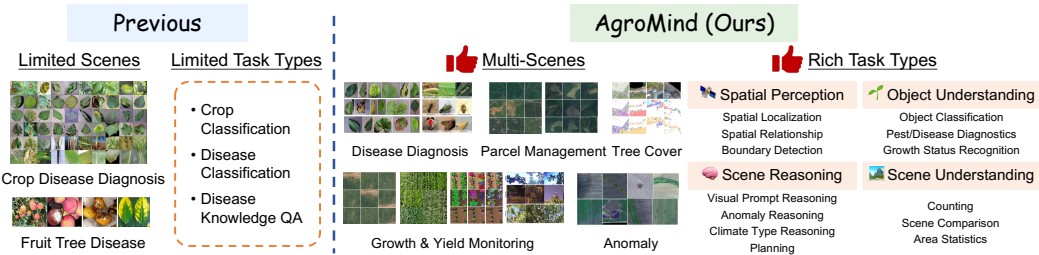

Figure 2: Comparison between AgroMind and previous works. Previous works focused on narrow tasks with limited data and task types. The AgroMind covers four hierarchical task dimensions, offering a more comprehensive evaluation framework through the integration of multiple data sources.

To begin with, current benchmarks suffer from insufficient scene diversity in terms of crop types, geographic coverage, and climatic conditions, limiting their ability to reflect the complexity and variability inherent in real-world agricultural environments. Existing datasets in the agriculture field are often limited in scope, primarily focusing on a single crop type or a specific scenario, such as crop diseases [20] or fruit diseases [21], which poses challenges in evaluating model performance in more dynamic and diverse agricultural scenarios (see Table 1). To achieve accurate and comprehensive benchmarking, it is imperative to develop datasets that cover a wide array of crops, diseases, pests, and environmental conditions, addressing the needs of various agricultural applications.

Furthermore, existing benchmarks for agricultural vision tasks often rely on oversimplified task design such as basic classification or detection [20, 22, 23], which fails to capture the complexity of real-world agricultural scenarios. In practice, agricultural applications demand models capable of spatial reasoning, contextual understanding, and support decision-making for tasks like yield forecasting or precision fertilization. While AgroBench[24] introduces a hierarchical task structure that aligns better with practical needs, the lack of quantitative evaluation metrics prevents it from enabling systematic performance assessment of LMMs in agriculture.

To address these gaps, we introduce **AgroMind**, a multi-scenario, multi-task benchmark designed to comprehensively evaluate LMMs in the agricultural RS environment. We integrate 9 public datasets and 1 private global parcel dataset to create a multi-scene dataset supporting various critical agricultural applications. AgroMind covers four core task dimensions: spatial perception, object understanding, scene understanding, and scene reasoning, with a total of 13 task types, ranging from crop identification and health monitoring to environmental analysis. AgroMind includes both basic tasks from previous benchmarks as well as reasoning tasks in agricultural real-world scenarios.

Table 1: Comparison between existing RS benchmarks and ours. Answer types include JD (Judgement), CO (Counting), TC (Text Choice), IC (Image Choice), OE (Open-ended), and BX (Bounding Box). Sensor types include UAV (Unmanned Aerial Vehicle), SL (Satellite), and CM (Camera).

| Dataset | Category | Size | Answer Type | | | | | | Sensors | | | Difficulty Level |
|---|---|---|---|---|---|---|---|---|---|---|---|---|
| | | | JD | CO | TC | IC | OE | BX | UAV | SL | CM | |
| RSIEval [25] | 10 types | 1K | ✓ | ✓ | | | ✓ | | | ✓ | | |
| VRSBench [19] | 10 types | 12K | ✓ | ✓ | | | ✓ | ✓ | ✓ | ✓ | | |
| EarthVQA [26] | 6 types | 21K | ✓ | ✓ | | | ✓ | | | ✓ | | ✓ |
| Urbench [17] | 14 types | 11.6K | ✓ | ✓ | ✓ | ✓ | ✓ | ✓ | | ✓ | ✓ | |
| LHRS-Bench [27] | 11 types | >1.1M | ✓ | ✓ | | | ✓ | ✓ | | ✓ | | |
| XLRS-Bench [18] | 16 types | 32K | ✓ | ✓ | ✓ | | ✓ | ✓ | | ✓ | | |
| CDDM [20] | - | 1M | ✓ | | | | ✓ | | | | ✓ | |
| AgriBench [24] | 17 types | >7K | | ✓ | | ✓ | ✓ | | | | ✓ | ✓ |
| Agri-LLaVA [22] | - | 400K | ✓ | | | | ✓ | | | | ✓ | |
| AgroInstruct [28] | 6 types | 70K | ✓ | | | | ✓ | | | | ✓ | ✓ |
| AgMMU [23] | 5 types | 5K | | | ✓ | | ✓ | | | | ✓ | |
| AgroMind (ours) | 13 types | 28K | ✓ | ✓ | ✓ | ✓ | ✓ | ✓ | ✓ | ✓ | ✓ | ✓ |

In summary, the contributions of this work can be concluded as:

• We construct a large-scale multi-modal and multi-scene agricultural remote sensing dataset, which includes 9 public datasets and 1 private global parcel dataset, containing 20,850 images and 28,482 QA pairs. The dataset spans critical applications such as pest and disease monitoring, parcel analysis, multiple crop recognition, as well as anomaly detection in agricultural scenarios.

• We propose AgroMind, a comprehensive multi-dimensional benchmark that assesses large models across 13 tasks under 4 key dimensions: spatial perception, object understanding, scene understanding, and scene reasoning, aiming to provide a systematic evaluation of model capabilities in agriculture.

• We conduct a comprehensive evaluation of 20 open-source and 4 closed-source LMMs on the proposed AgroMind. Experimental results reveal the limitations of current LMMs in agricultural remote sensing applications, and present the future directions for expert knowledge-based LMMs.

## 2 Related work

### 2.1 Large Multimodal Models

Leveraging the advanced language reasoning and understanding capabilities of Large Language Models (LLMs) [13, 29], LMMs have demonstrated powerful capabilities in multimodal tasks. Recent advancements in LMMs have led to the development of both closed-source systems, such as GPT-4o [11] and Gemini [12], and open-source frameworks like the InternVL series [30], LLaVA series [31, 32], and VILA series [33]. However, despite their success in general tasks, LMMs still face challenges when dealing with domain-specific tasks.

Recent years have seen growing interest in applying LMMs to remote sensing [34, 35], leveraging their ability to integrate visual and textual data for intelligent decision-making. Notable examples include GeoChat [36] and EarthGPT [37], which demonstrate cross-modal capabilities in geospatial analysis. Models such as Agri-LLaVA [22], AgroGPT [28], and WheatRustVQA [38] demonstrate the potential in agricultural tasks like disease diagnosis and crop yield prediction. However, these works primarily focus on narrow scope tasks (e.g., crop classification, disease detection) and often ignore the complexity of agricultural scenarios, such as geospatial reasoning and scene understanding. Our objective is to assess LMMs in a wider variety of agricultural RS tasks, broadening the scope to include more extensive view coverage and fully harnessing their potential in agricultural scenarios.

### 2.2 General multimodal benchmark

With the rapid advancement of LMMs, traditional benchmarks such as captioning [39], VQA [40, 41], GQA [42], and TextVQA [43] have become insufficient for evaluating their comprehensive capabilities. Pioneering works such as MME and MMBench established multidimensional evaluation protocols encompassing perception, reasoning, and social cognition through thousands of carefully

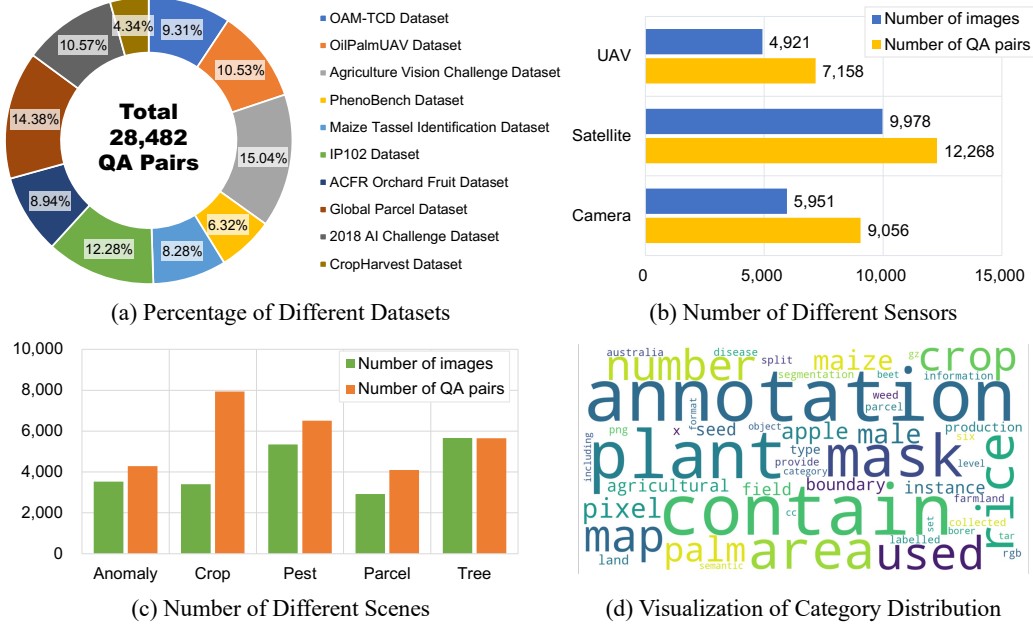

(a) Percentage of Different Datasets

(b) Number of Different Sensors

(c) Number of Different Scenes

(d) Visualization of Category Distribution

Figure 3: Dataset statistics from different aspects.

curated questions. The introduction of MMMU [44] further elevated assessment rigor by incorporating expert-level knowledge evaluation, exposing fundamental limitations in current models' reasoning capacities. Recently, benchmarks like Seed-Bench [45], MMTBench [46], and MME-Realworld [47] establish large-scale real-world evaluation frameworks. Given the inadequacy of current benchmarks in addressing the unique challenges of specialized fields, there is an urgent need to develop LMM benchmarks that are explicitly tailored to domain-specific knowledge.

### 2.3 Remote sensing multimodal benchmark

The RS community has seen a surge in initiatives dedicated to assessing LMMs performance. Multimodal benchmarks have rapidly evolved from early perception-oriented tasks such as image captioning and object classification to more complex reasoning tasks including visual question answering, spatial reasoning, and object relationship understanding. Pioneering works such as EarthVQA [26], RSIEval [25], LHRS-Bench [27], VRSBench [19], UrBench [17], and XLRS-Bench [18] have significantly advanced the evaluation of large multimodal models (LMMs) in general and urban remote sensing scenarios. These efforts have significantly enriched the evaluation of LMMs in general and urban remote sensing contexts.

However, these benchmarks are largely limited to urban, disaster response, or environmental domains, providing insufficient coverage of agricultural remote sensing tasks such as crop identification, growth monitoring, and pest or disease diagnosis. While early efforts like CDDMBench [20] and AgriBench [24] have explored multimodal evaluation in agriculture, they remain constrained by narrow task coverage, limited scene diversity, and small-scale data. Therefore, it is crucial to construct a comprehensive multi-dimensional benchmark for real agricultural RS tasks, to promote the application of LMMs in agricultural intelligence.

## 3 Dataset

### 3.1 Dataset statistics

This section presents fundamental statistical characteristics of the AgroMind dataset. The dataset is constructed by integrating nine publicly available RS datasets and one proprietary global parcel dataset, specifically including OAM-TCD [48], OilPalmUAV [49], AVC [50], PhenoBench [51], Maize Tassel Identification Dataset (Iflytek AI Competition 2024), IP102 [52], ACFR [53], 2018 AI

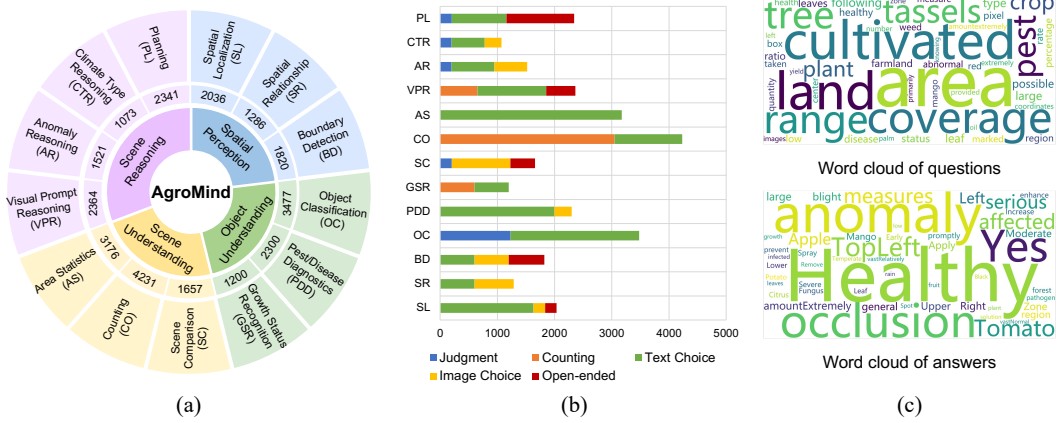

Figure 4: Statistical information of AgroMind.

Challenge Dataset, CropHarvest[54], and Global Parcel Dataset. Through a selection process, we retained 20,850 images and generated 28,482 question-answer (QA) pairs from these visual data, ensuring comprehensive diversity across agricultural scenarios. The proportional of different source datasets are quantitatively illustrated in Fig. 3 (a). More details can be found in Appendix B.1.

**Statistics for different sensors.** The AgroMind dataset was compiled from three sensor types: unmanned aerial vehicles (UAVs), satellite platforms, and cameras. Fig. 3 (b) presents the exact numerical distribution of images and corresponding QA pairs for each sensor type, revealing a QA pair ratio of approximately 9:12:7 for ground-based cameras, satellite platforms, and UAVs, respectively. Notably, while satellite and ground imagery constitute the majority of the dataset, UAV-derived annotations still encompass 7,000 QA pairs, ensuring sufficient coverage to validate model robustness in multi-perspective agricultural tasks.

**Statistics for different scenes.** Fig. 3 (c) demonstrates the comprehensive coverage of agricultural scenarios spanning five critical categories: Anomaly Detection, Crop Monitoring, Pest Identification, Parcel Delineation, and Tree Analysis. The Crop Monitoring category contains approximately 3,500 high-quality images paired with 8,000 annotated QA pairs, forming the largest data subset. The Pest Identification and Tree Analysis categories consist of 5,000 images (with 6,000 QA pairs) and 6,000 images (with 6,000 QA pairs), respectively. Notably, the inclusion of specialized scenarios such as anomaly detection and parcel delineation demonstrates the capacity to address emerging challenges in precision agriculture.

**Visualization of category distribution.** The dataset encompasses a wide range of agricultural and geospatial categories, as demonstrated in Fig. 3 (d). Critical semantic classes include "annotation" for precise farmland boundary delineation and "mask" for vegetation segmentation, while domain-specific categories such as "plant", "rice", "crop", and "palm" underscore the dataset's focus on targeted agricultural analysis. The visualization quantitatively confirms balanced representation between common agricultural scenarios (e.g., field-level annotations) and specialized applications (e.g., pixel-wise control mechanisms).

**Geospatial distribution.** Our AgroMind dataset encompasses 106 geographic regions globally, including major agricultural areas like China, the United States, and the European Union. This extensive coverage guarantees significant diversity in key aspects such as climate regimes, land-use intensity, and crop phenology, establishing it as one of the foremost benchmarks in agricultural remote sensing for its spatial scale. For a more detailed breakdown and analysis of the data's geographic distribution, please refer to Appendix B.1.

**Temporal and phenological coverage.** AgroMind features rich, multi-scale temporal information. It captures the complete phenological stages of crops from growth to maturity, identifies time-sensitive anomalies like waterlogging, and covers multi-seasonal land changes through multi-year satellite imagery. This supports the evaluation of models' long-term reasoning capabilities, such as for yield prediction. More details are available in Appendix B.1.

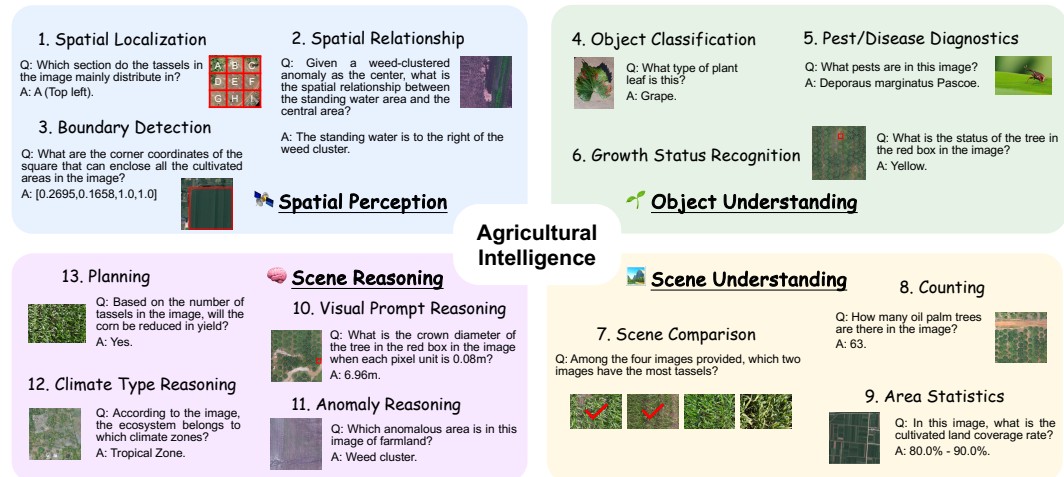

Figure 5: Hierarchical task system of AgroMind. Based on the progressive cognitive pipeline in agricultural intelligence, AgroMind constructs four evaluation dimensions, covering 13 different task types, and comprehensively evaluates the performance of LMMs in agricultural scenarios.

## 3.2 Dataset preprocessing

Customized processing protocols were applied to heterogeneous data sources: (1) TIFF-format remote sensing images were processed based on content: Colorful images recognizable to the human eye were converted to PNG/JPG formats to ensure model compatibility and prevent parsing failures; while images containing primarily non-visible spectral bands had all bands converted into grayscale blocks that were concatenated, preserving the original spectral information through grayscale representation; (2) High-resolution geospatial images underwent randomized parcel-boundary cropping to generate multi-scale samples simulating spatial query variations; (3) A manual screening step was performed on augmented images to remove samples exhibiting artifacts such as content overlap or excessive flipping that could distort semantic information;(4) Instance-level statistical analysis and bounding box annotation were performed using original labels and masks. The resulting standardized dataset contains hierarchical annotations across three levels: pixel-level (segmentation masks), instance-level (object detection boxes), and parcel-level (agricultural boundaries). Original image resolutions were preserved (300×300 to 4,500×4,500 pixels) to maintain data integrity. Moreover, we applied a logic enhancement strategy in the QA pair design, artificially predefining the hierarchical reasoning process instead of directly using the raw annotations as the answers of QA pairs. Complete collection procedures and statistical details are documented in the Appendix B.

## 4 AgroMind benchmark

We introduce AgroMind, a comprehensive multimodal benchmark designed to assess the capabilities of LMMs in agricultural scenarios, containing 28,482 image-question-answer pairs. The AgroMind is characterized by the following features: **(1) Broad task coverage.** AgroMind covers four categories of tasks (Fig. 4(a)): spatial perception, object understanding, scene understanding and reasoning, and a total of 13 task types, covering typical visual language requirements in agricultural production. **(2) versatile evaluation.** It supports a wide range of question formats such as judgment, counting, image choice, text choice, and open-ended questions, allowing for comprehensive evaluation from basic perception to high-level reasoning (Fig. 4(b)). **(3) Domain-specific focus.** As visualized in the word clouds, the benchmark emphasizes agricultural-specific terms such as cultivated, land, pest, healthy and anomaly, demonstrating its alignment of the domain (Fig. 4(c)).

### 4.1 Evaluation dimensions

AgroMind comprehensively evaluates LMMs in agricultural scenarios from 4 evaluation dimensions. Fig.5 illustrates the specific task types under each evaluation dimension. Please refer to the appendix

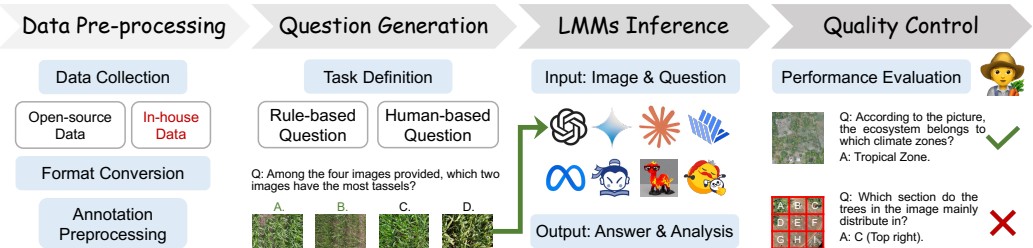

Figure 6: The benchmark curation pipeline contains four stages, i.e., data pre-processing, question generation, LMMs Inference, and quality control.

for additional task questions. The hierarchical evaluation system reflects the ability from "seeing clearly" to "reasoning accurately", aligning with practical demands in agricultural intelligence.

**Spatial perception** forms the foundation of agricultural scene understanding by assessing the ability to extract and reason about spatial information from images (Task 1-3 in Fig. 5). Key tasks at this level include Spatial Localization (SL), e.g., identifying where tassels are mainly distributed in a field; Spatial Relationship (SR), e.g., determining the relative position of standing water to a weed cluster; and Boundary Detection (BD), e.g., predicting the coordinates enclosing cultivated areas.

**Object understanding** targets semantic recognition at the object level, evaluating the capability to identify specific agricultural entities and their attributes (Task 4-6 in Fig. 5). This includes Object Classification (OC), e.g., identifying a plant leaf as grape, Pest/Disease Diagnostics (PDD), e.g., recognizing pest species such as Deporaus marginatu, and Growth Status Recognition (GSR), e.g., determining whether a tree appears yellow or healthy.

**Scene understanding** moves beyond individual objects to holistic interpretation, requiring models to aggregate and reason across multiple regions or images (Task 7-9 in Fig. 5). Typical tasks include Scene Comparison (SC), e.g., identifying images with the most tassels from a set of samples; Counting (CO), e.g., estimating the number of oil palm trees in an image; and Area Statistics (AS), e.g., calculating cultivated land coverage rates.

**Scene reasoning** represents the most advanced level, emphasizing the capacity for visual-grounded inference and cross-domain knowledge application (Task 10-13 in Fig. 5). Tasks at this dimension include Visual Prompt Reasoning (VPR), e.g., inferring tree crown diameter based on image resolution; Anomaly Reasoning (AR), e.g., identifying anomalous weed-cluster regions, Climate Type Reasoning (CTR), e.g., determining climate zone from ecosystem features, and Planning (PL), e.g., predicting yield reduction.

## 4.2 Benchmark curation

The pipeline of AgroMind covers four key steps from data preparation to quality control, ensuring accurate measurement of LMMs performance in agricultural RS tasks, as detailed in Fig. 6. More details can be found in Appendix B and C.

**Data pre-processing** focuses on data collection and organization. It involves gathering both open-source data and in-house data. After collection, format conversion is carried out to make the data suitable for subsequent processing. Meanwhile, annotation pre-processing is performed, laying the foundation for the following tasks.

**Question generation** are based on task definition. There are two ways to generate questions: rule-based question and human-based question. Rule-based questions are characterized by their normativity and logic, Human-based questions, on the other hand, are more flexible and diverse, capable of exploring data information from various perspectives.

**LMMs inference** takes the preprocessed image with the generated question as input to a large multimodal model. After analysis processing, the model outputs the answer and analysis. Through the inference ability of the models, questions related to agricultural RS are answered and analyzed.

**Quality control** stage mainly evaluates the output of the models. Each answer generated by LMMs is systematically compared with the standard answers annotated by human experts. If the answer

Table 2: Performance comparison across multiple tasks and models. The overall score represents the average accuracy across all tasks. Task names are abbreviated for brevity.

| Model | Spatial Perception | | | Object Understanding | | | Scene Understanding | | | Scene Reasoning | | | | Overall |
|---|---|---|---|---|---|---|---|---|---|---|---|---|---|---|
| | SL | SR | BD | OC | PDD | GSR | SC | CO | AS | VPR | AR | CTR | PL | |
| Human | 19.15 | 15.00 | 40.83 | 57.60 | 46.17 | 47.22 | 35.49 | 28.88 | 39.46 | 12.33 | 25.62 | 34.07 | 13.50 | 33.15 |
| Random | 18.03 | 17.97 | 16.48 | 35.65 | 25.65 | 10.00 | 24.39 | 4.99 | 21.20 | 9.19 | 19.87 | 21.70 | 26.96 | 19.24 |
| GPT-4o [11] | 41.38 | 35.16 | 33.52 | **78.55** | **66.09** | 55.83 | 43.90 | 23.75 | 46.20 | 16.10 | 27.15 | **69.81** | 30.90 | **43.14** |
| Gemini-1.5-Flash [12] | 29.56 | **39.06** | 24.73 | 77.68 | 62.17 | 55.83 | 33.54 | 23.75 | 43.04 | 22.88 | 29.14 | 68.87 | 27.90 | 40.92 |
| Gemini-1.5-Pro [12] | 27.59 | 23.44 | 24.73 | 77.39 | 52.61 | 45.83 | 39.02 | **32.30** | 47.78 | 19.92 | **34.44** | 66.04 | 30.04 | 41.06 |
| Claude-3.5-Sonnet [55] | 24.14 | 17.19 | 40.11 | 57.97 | 50.00 | 57.50 | **44.51** | 24.47 | 35.44 | 20.34 | 33.11 | 50.94 | 36.05 | 37.11 |
| DeepSeek-VL2-tiny [62] | 16.75 | 24.22 | 19.78 | 59.42 | 25.22 | 55.83 | 29.88 | 21.38 | 14.87 | 17.37 | 15.89 | 44.34 | 21.89 | 27.51 |
| DeepSeek-VL2-small [62] | 16.26 | 20.31 | 24.18 | 61.45 | 28.70 | **65.83** | 31.71 | 19.71 | 20.57 | 19.07 | 19.87 | 38.68 | 8.58 | 28.08 |
| TinyLLaVA [56] | 29.56 | 17.97 | 24.73 | 68.41 | 23.48 | 30.00 | 24.39 | 23.99 | 31.65 | 3.39 | 13.91 | 45.28 | 9.44 | 28.01 |
| InternVL2-2B [30] | **53.69** | 19.53 | 17.03 | 41.16 | 30.43 | 42.50 | 26.22 | 21.62 | 29.11 | 14.83 | 27.15 | 39.62 | 14.16 | 28.40 |
| InternVL2-4B [30] | 29.06 | 23.44 | 25.27 | 48.70 | 44.78 | 19.17 | 30.49 | 19.00 | 46.84 | 19.49 | 30.46 | 50.94 | 12.88 | 31.15 |
| XComposer2-4KHD [57] | 29.06 | 27.34 | 32.97 | 51.88 | 40.00 | 44.17 | 34.15 | 22.33 | 27.53 | 11.86 | 37.75 | 43.40 | **38.63** | 33.02 |
| LLaVA-NeXT-7B-Mistral [32] | 33.50 | 22.66 | 40.11 | 44.64 | 40.00 | 36.67 | 21.95 | 24.70 | 36.71 | 8.05 | 15.23 | 39.62 | 11.59 | 29.17 |
| LLaVA-NeXT-7B-Vicuna [32] | 4.43 | 22.66 | 40.11 | 52.75 | 30.43 | 22.50 | 20.12 | 20.90 | 21.84 | 14.41 | 11.26 | 40.57 | 24.89 | 25.82 |
| InstructBLIP-Vicuna-7B [58] | 9.85 | 16.41 | 20.88 | 33.33 | 21.74 | 10.83 | 19.51 | 10.93 | 21.52 | 4.66 | 20.53 | 20.75 | 11.16 | 17.39 |
| LLaVA-NeXT-Interleave-7B [31] | 12.32 | 15.62 | 22.53 | 32.46 | 20.87 | 4.17 | 21.95 | 10.69 | 20.25 | 13.98 | 18.54 | 29.25 | 21.89 | 19.01 |
| Mantis-LLaMA3-SigLIP [63] | 23.65 | 21.09 | 25.27 | 65.51 | 32.17 | 12.50 | 29.88 | 26.84 | 33.54 | 12.29 | 21.19 | 46.23 | 30.04 | 31.18 |
| Mantis-Idefics2 [63] | 16.75 | 20.31 | 31.32 | 63.77 | 44.35 | 37.50 | 22.56 | 24.23 | 20.89 | 8.47 | 13.91 | 56.60 | 30.90 | 30.41 |
| LLaVA-NeXT-8B [32] | 35.47 | 16.41 | **41.76** | 57.68 | 28.70 | 30.00 | 19.51 | 26.84 | 34.81 | 10.59 | 17.22 | 37.74 | 33.48 | 31.53 |
| InternVL2-8B [30] | 24.63 | 23.44 | 19.23 | 72.17 | 43.48 | 24.17 | 32.93 | 23.75 | **54.11** | 22.46 | 18.54 | 61.32 | 27.90 | 36.30 |
| Idefics-2-8b [59] | 21.67 | 14.84 | 26.37 | 58.84 | 33.91 | 32.50 | 21.95 | 23.04 | 19.94 | 13.98 | 22.52 | 49.06 | 9.44 | 27.09 |
| LLaVA-NeXT-13B [32] | 45.32 | 19.53 | 36.81 | 51.88 | 33.04 | 31.67 | 23.17 | 22.09 | 34.81 | 5.51 | 21.85 | 42.45 | 26.18 | 30.69 |
| InternVL2-26B [30] | 46.31 | 24.22 | 19.78 | 50.72 | 42.61 | 55.00 | 39.02 | 24.47 | 44.30 | **33.05** | 24.50 | 56.60 | 31.76 | 37.25 |
| LLaVA-NeXT-34B [32] | 49.26 | 16.41 | 39.56 | 59.42 | 40.00 | 30.83 | 25.61 | 27.32 | 43.35 | 17.80 | 19.87 | 45.28 | 28.33 | 35.52 |
| GeoChat [60] | 14.78 | 19.53 | 23.63 | 48.70 | 34.78 | 30.83 | 16.46 | 19.48 | 20.57 | 8.47 | 29.80 | 51.89 | 24.89 | 25.93 |
| GeoLLaVA-8K [61] | 9.85 | 25.78 | 17.58 | 34.78 | 15.22 | 7.50 | 17.68 | 4.75 | 16.14 | 7.63 | 14.57 | 34.91 | 8.58 | 15.73 |

does not match the facts, has logical holes, or omits key information, it is marked as false. Through performance evaluation, it is determined whether the answers given by the models are correct.

## 5 Experiments

In this section, we evaluate various LMMs on our proposed AgroMind. We consider close-sourced models, open-sourced single-image models and open-sourced multi-image models, and perform evaluations under zero-shot settings. In the following sections, we first introduce our evaluated models and evaluation protocols. Then, we present the main results, highlighting challenges, and performance across different tasks and difficulty levels. Finally, we provide a detailed analysis of model size effects and compare GPT-4o with human performance.

### 5.1 Evaluation setups

**Evaluated Models.** We evaluate a total of 24 LMMs using the AgroMind benchmark, including 4 close-sourced and 20 open-sourced models. The close-sourced models including GPT-4o [11], Gemini 1.5 Flash [12], Gemini 1.5 Pro [12], and Claude 3.5 Sonnet [55] are accessed via their official APIs. The open-sourced models fall into two categories: single-image models, including TinyLLaVA [56], the LLaVA-NeXT series [32], Xcomposer2-4KHD [57], InstructBLIP-Vicuna-7B [58], Idefics-2-8B [59], GeoChat [60], and GeoLLaVA-8K [61]; and multi-image models, including the DeepSeek-VL2 series [62], InternVL2 series [30], LLaVA-NeXT-Interleave-7B [31], the Mantis series [63], and Idefics-2-8B [59] (see Appendix C.2 for details). All close-sourced models run on machines equipped with NVIDIA A800 GPUs.

**Human users.** We recruited approximately twenty participants, comprising undergraduate and graduate students, ensuring that each question was independently answered by at least three volunteers. We report the average accuracy across all participants as the final human performance. In addition to LLM-based models, we also include the Random baseline, which selects an answer at random from the candidate options for each question.

**Evaluation protocols.** The AgroMind benchmark comprises five question types: judgment, counting, text choice, image choice and open-ended questions. We follow standard evaluation setups from MMMU [44] and UrBench [17] to parse and assess the answers generated by LMMs. For all question types except open-ended ones, we adopt strict string matching, the answer can be marked correct only if it matches the ground truth exactly. For open-ended questions, correctness is determined based on the semantic similarity between the model-generated answers and the reference answers (see Appendix C.4 for details). Additionally, to ensure that models are evaluated solely based on their own visual understanding rather than leveraging multi-image input advantages, all multi-image questions are converted into single-image input by concatenating all related images into one.

## 5.2 Main results

**Overall challenge presented in AgroMind.** As indicated in Table 2, AgroMind poses significant challenges to current SOTA LMMs. Although some models, such as GPT-4o, have surpassed human performance, 17 out of the 24 evaluated models still underperform compared to humans. Both close-sourced and open-sourced models fall short of consistently outperforming humans on all tasks. The challenging nature of our benchmark indicates that current LMMs' strong performance on the standard tasks are not generalized to the multi-modal and multi-scene agricultural scenarios. Additionally, domain-specific models like GeoChat [60] and GeoLLaVA-8K [61], while effective in general remote sensing tasks, demonstrate limitations in complex agricultural scenarios.

Fig. 7 illustrates the performance of several mainstream LMMs across 13 distinct agricultural tasks, highlighting their overall capability distribution in multi-task agricultural settings. The radar chart reveals a significant variation in performance across tasks, suggesting heterogeneous strengths and structural limitations among the models. A clear trend emerges: the models generally perform better on perceptual tasks such as vegetation recognition, disease identification, and ecological type classification, while struggling on reasoning and quantitative tasks like area estimation, object counting, and spatial localization.

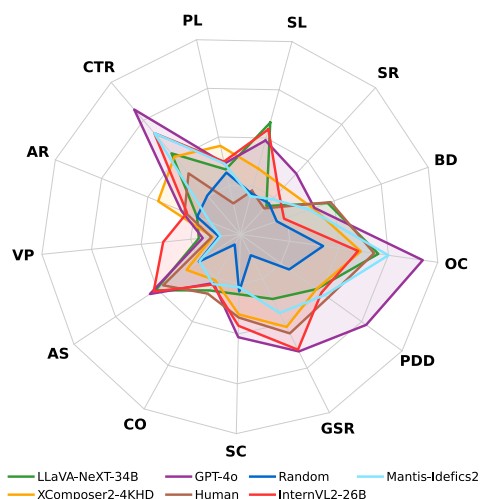

Figure 7: Performance of 5 leading LMMs, as well as that of the human and random guess on AgroMind.

**LMMs' performances across task dimensions.** Table 2 summarizes the performance of various LMMs across 13 tasks. Most LMMs perform poorly on Spatial Perception tasks such as BD, indicating difficulties in analyzing the spatial distribution of crops or farmland. While LMMs achieve relatively high results on Object Understanding tasks like OC and PDD, their performance drops significantly on GSR, which is crucial for crop health monitoring. For Scene Understanding, most LMMs underperform compared to humans, indicating that they tend to reproduce answers seen in their training data rather than understand the agricultural scene itself. Interestingly, LMMs excel at Scene Reasoning tasks like CTR and PL by using their broad knowledge base to give expert answers. However, they perform much worse than humans on VPR, which requires visual reasoning, indicating that LMMs still need significant improvement for vision-intensive agricultural tasks.

**Model performance across different levels.** To better analyze model performance, we report the performance of both humans and various LMMs across three difficulty levels, as shown in Table 3. The definitions of these levels are provided in Appendix C.3. We find that most LMMs outperforms humans on hard questions, likely due to the fact that our volunteers are primarily students who may lack specialized knowledge required for certain questions. However, the experimental results also reveals shortcomings of LMMs on easy and medium-level questions, which are more commonly encountered in real-life agricultural scenarios.

**Is model size directly linked to performance?** Fig. 8 illustrates the performance of a series of models with varying parameter scales. While larger models generally tend to achieve better

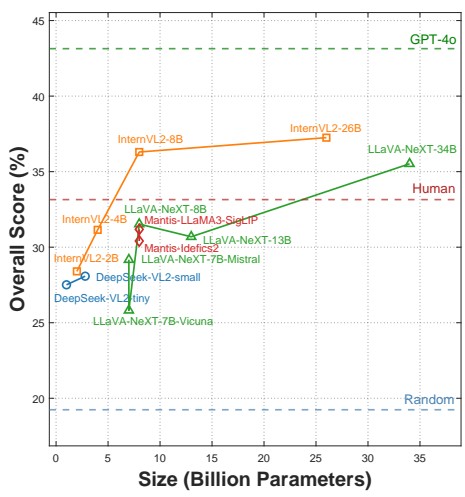

Figure 8: Model performance under different scales, with human responses and random answers as references.

Table 3: Model performance across difficulty levels.

| Model | Easy | Medium | Hard | Overall |
|---|---|---|---|---|
| Human | 78.01 | 45.06 | 13.36 | 33.15 |
| Random | 23.21 | 25.84 | 11.41 | 19.24 |
| GPT-4o [11] | **78.16** | **55.77** | **24.29** | **43.14** |
| Gemini-1.5-Flash [12] | 77.47 | 53.68 | 21.62 | 40.92 |
| Gemini-1.5-Pro [12] | 75.09 | 53.68 | 22.44 | 41.06 |
| Claude-3.5-Sonnet [55] | 66.21 | 51.09 | 18.35 | 37.11 |
| DeepSeek-VL2-tiny [62] | 72.35 | 31.02 | 14.64 | 27.51 |
| DeepSeek-VL2-small [62] | 70.31 | 33.03 | 14.49 | 28.08 |
| TinyLLaVA [56] | 60.41 | 35.12 | 14.64 | 28.01 |
| InternVL2-2B [30] | 55.63 | 31.10 | 20.06 | 28.40 |
| InternVL2-4B [30] | 54.27 | 42.06 | 16.42 | 31.15 |
| XComposer2-4KHD [57] | 61.09 | 40.47 | 20.28 | 33.02 |
| LLaVA-NeXT-7B-Mistral [32] | 57.68 | 37.37 | 15.68 | 29.17 |
| LLaVA-NeXT-7B-Vicuna [32] | 49.83 | 32.78 | 14.41 | 25.82 |
| InstructBLIP-Vicuna-7B [58] | 34.47 | 20.57 | 10.85 | 17.39 |
| LLaVA-NeXT-Interleave-7B [31] | 28.67 | 23.33 | 13.08 | 19.01 |
| Mantis-LLaMA3-SigLIP [63] | 54.61 | 43.06 | 15.53 | 31.18 |
| Mantis-Idefics2 [63] | 61.43 | 42.14 | 13.22 | 30.41 |
| LLaVA-NeXT-8B [32] | 57.34 | 42.06 | 16.57 | 31.53 |
| InternVL2-8B [30] | 64.51 | 49.41 | 18.50 | 36.30 |
| Idefics-2-8b [59] | 57.68 | 33.86 | 14.41 | 27.09 |
| LLaVA-NeXT-13B [32] | 56.31 | 39.30 | 17.46 | 30.69 |
| InternVL2-26B [30] | 67.58 | 46.15 | 22.73 | 37.25 |
| LLaVA-NeXT-34B [32] | 59.04 | 46.40 | 20.73 | 35.52 |
| GeoChat [60] | 51.54 | 33.28 | 13.82 | 25.93 |
| GeoLLaVA-8K [61] | 24.57 | 22.58 | 7.73 | 15.73 |

performance, the overall performance is more influenced by the model's architecture rather than its parameter size. For instance, the LLaVA-NeXT model with 34B parameters performs worse than the InternVL2 model with 26B parameters. Furthermore, when the parameter size of LLaVA-NeXT increases from 8B to 13B, its accuracy even drops by 0.84%.

**Does GPT-4o exceed human performance?** As shown in Table 2, GPT-4o's overall accuracy surpasses human performance by 9.99%. We attribute this to the fact that our human participants are primarily students, who may lack the specialized knowledge included in GPT-4o's extensive knowledge base and GPT-4o is the most SOTA model, which demonstrates a substantial lead over all other evaluated models. However, despite outperforming humans in most tasks, GPT-4o falls far behind in tasks such as BD and CO, which exposes its limitations in spatial perception and counting ability in agriculture. More results can be found in Appendix C.5.

## 6 Conclusion

In this work, we present **AgroMind**, a comprehensive benchmark tailored for evaluating LMMs in the context of agricultural remote sensing. The benchmark is constructed from real-world data sources and comprises 28,482 question-answer pairs across 13 task types and 4 progressive cognitive levels: spatial perception, object understanding, scene understanding, and scene reasoning. All tasks are designed to reflect practical agricultural needs, enabling a systematic assessment of LMMs' ability to interpret, infer, and reason over complex agricultural scenes. We conduct thorough evaluations of 24 representative LMMs, revealing notable performance gaps between current models and expert-level understanding in agriculture-related scenarios. AgroMind not only highlights the limitations of existing models but also provides a new foundation for advancing domain-specific multimodal intelligence. We hope this work inspires future research toward developing more capable, generalizable, and agriculture-aware large models.

## 7 Acknowledgements

This work was supported in part by the Guangdong Science and Technology Program under Grant 2024B0101040005; in part by the National Natural Science Foundation of China under Grant T2125006 and Grant 42401415; in part by Shenzhen Science and Technology Program under Grant KCXFZ20240903093759004 and Grant KJZD20230923115106012; in part by the Fundamental Research Funds for the Central Universities, Sun Yat-sen University, under Project 24xkjc002; and in part by Jiangsu Innovation Capacity Building Program under Project BM2022028.

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

# Can Large Multimodal Models Understand Agricultural Scenes? Benchmarking with AgroMind
## (Supplementary material)

## Table of Contents in Appendix

# A Broader impacts

The deployment of LMMs in agriculture holds tremendous promise for enhancing global food security, environmental sustainability, and rural economies. However, the lack of standardized evaluation frameworks tailored for the unique complexities of agricultural RS has impeded both academic progress and practical deployment. By introducing AgroMind, we aim to lay the groundwork for a systematic and scalable way to assess the real-world applicability of LMMs in agriculture.

By constructing a comprehensive benchmark that spans four critical dimensions: Spatial Perception, Object Understanding, Scene Understanding, and Scene Reasoning, AgroMind not only addresses long-standing gaps in scene diversity and task complexity, but also offers a standardized platform for evaluating model capabilities in complex agricultural contexts. AgroMind goes beyond traditional tasks such as crop classification or object detection, and instead introduces more challenging tasks, such as spatial logical reasoning, fine-grained semantic judgment, and multi-object association, that better reflect the complexity of real-world agricultural scenarios.

AgroMind has the potential to reshape the way we assess intelligent systems in agricultural monitoring, fostering progress in high-impact areas such as crop health detection, precision farming, and environmental sustainability. Furthermore, the experiment exposes the current limitations of LMMs when applied to domain-specific knowledge and real-world spatial reasoning, encouraging further research on model robustness, domain adaptation, and interpretability.

In a broader context, AgroMind can serve as a foundational tool for policymakers, agricultural technologists, and researchers aiming to deploy AI in real-world agricultural decision-making systems. It provides a rigorous evaluation basis for selecting trustworthy models and sets a precedent for future multimodal benchmarks in other mission-critical verticals such as climate monitoring, biodiversity, and disaster response.

# B Dataset description

## B.1 Data collection

Existing agricultural remote sensing benchmarks predominantly focus on single spatial resolutions, homogeneous data modalities, homogeneous QA templates, or isolated scenarios, resulting in limited scene coverage and overly simplified task designs. To bridge this gap, we systematically integrate nine public datasets with one proprietary field-parcel dataset to establish a comprehensive "Sky–Air–Ground" agricultural visual-QA benchmark. Details regarding each dataset's acquisition year, key attributes, resolution, and source are provided in Table 4. The selected datasets align with key agricultural decision pathways and span five core domains: (1) crop phenology monitoring, (2) pest and disease detection, (3) composite anomaly diagnosis via multi-sensor fusion, (4) precision parcel management through field delineation, and (5) high-resolution forestry area assessment(see Fig 3 (c)). This multi-tiered coverage significantly extends the scenario breadth and task granularity of existing benchmarks in holistic agricultural assessment, thereby enabling comprehensive evaluation from parcel-level operations to ecosystem-level services. By leveraging heterogeneous data sources—satellite, UAV, and ground-based sensors(multispectral/hyperspectral), we further introduce a three-dimensional evaluation framework (Spatial–Object–Scene)(see Fig 3 (b)) to rigorously assess cross-modal feature fusion and agronomic reasoning. Unlike prior work, our benchmark enables joint evaluation at the parcel, farm, and regional scales and incorporates a dedicated forestry detection dimension, markedly enhancing the ecological-service assessment capabilities of agricultural RS models. The diverse, scenario-specific tasks encompassed by this benchmark (see Fig 10) lay the groundwork for robust and reproducible evaluation of large-scale multimodal models in precision agriculture.

**Geospatial distribution.** To address spatial coverage concerns, we conducted statistical analysis of global data distribution (see Fig. 9). AgroMind intentionally spans **106 geographic regions** including China, the United States, European Union nations, Brazil, Southeast Asia, and parts of Africa, reflecting agricultural diversity across crops, climates, and land-use patterns, achieving an extensive spatial coverage to date for agricultural remote sensing benchmarks.This global representation captures critical agricultural diversity across: (i) Climate regimes – from tropical monsoons (Indonesia) to Mediterranean (Italy) and continental (Ukraine); (ii) Land-use intensity – ranging from industrial monocultures (U.S. Corn Belt) to polycultures (Tanzanian farmlands); (iii) Crop

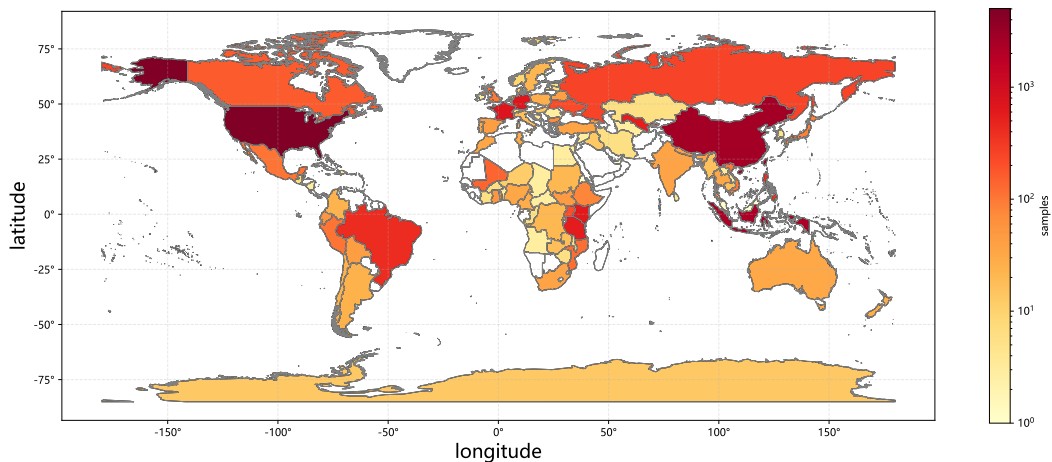

Figure 9: Geographical coverage map of datasets.

phenology – covering growth cycles of many staple crops (maize, rice, oil palm, etc.). The rich geographic diversity of the dataset allows robust cross-regional generalization validation and provides foundational resources for global agricultural sustainability research.

**Data scene diversity.** We carefully select ten datasets, including nine public datasets and one proprietary global parcel dataset, which cover multiple core scenarios and multimodal data characteristics in agricultural remote sensing. The OAM-TCD Dataset[48] and OilPalmUAV Dataset[49] provide high-resolution global canopy maps and oil-palm tree health assessments, respectively, allowing models to understand vegetation conditions at both macro and micro scales. In addition, the OAM-TCD Dataset[48] encompasses ecologically diverse regions from both the Northern and Southern Hemispheres (e.g., tropical rainforests and temperate woodlands), facilitating precise quantification of forest coverage and area for large-scale forestry monitoring. Meanwhile, the Agriculture Vision Challenge[50] and PhenoBench Dataset[51], by providing pixel-level annotations, emphasize in-field anomaly detection and detailed crop condition analysis, enhancing model capability in handling complex scene details. To support these tasks, the Agriculture Vision Challenge[50] combines RGB and near-infrared channels (512×512 pixels) with six binary masks (e.g., cloud shadow, double plant) to construct a multimodal anomaly diagnosis task. The PhenoBench Dataset[51] utilizes visibility maps (1024×1024 pixels) to quantify plant occlusion levels and frames tasks related to weed coexistence and plant integrity assessment. Furthermore, in the domain of fine-grained cropland management, our 7-Province Ground dataset integrates farmland vector data from multiple provinces with multispectral imagery, enabling precise field boundary delineation and dynamic coverage rate calculation, and the CropHarvest satellite dataset[54] provides multispectral and hyperspectral cropland classification, supporting detailed crop type identification. This supports tasks such as cropland area estimation and boundary recognition, significantly enhancing the practicality of models in agricultural planning and resource optimization.

**Multi-sensor heterogeneous data integration.** By integrating data from satellite, aerial, and ground-level sensors, this benchmark constructs a relatively comprehensive multi-source heterogeneous dataset for agricultural remote sensing. At the satellite level, the 7-Province Ground Dataset provides multispectral imagery and farmland vector data, supporting field boundary segmentation, area estimation, and coverage rate computation. At the aerial level, the OilPalmUAV[49] Dataset, with high-resolution canopy imagery, serves as the core for low-altitude plant counting and growth status monitoring. Ground-based sensor data further enrich the spectral diversity: PhenoBench Dataset[51] provides hyperspectral data for analyzing plant physiological conditions, while the 2018 AI Challenge Dataset offers disease-annotated leaf images (with 26 labeled disease types), which are paired with severity labels to generate plant disease reasoning QA tasks. To tackle cross-modal fusion challenges, the IP102 Dataset [52], which includes 102 insect categories with VOC-style annotations, is reformulated into a distractor-choice question format to test models' zero-shot pest identification ability. Through the coordinated use of multiple spatial resolutions, multiple spectral sources (RGB, multispectral, NIR), and diverse annotation formats (pixel-level masks, instance bounding boxes,

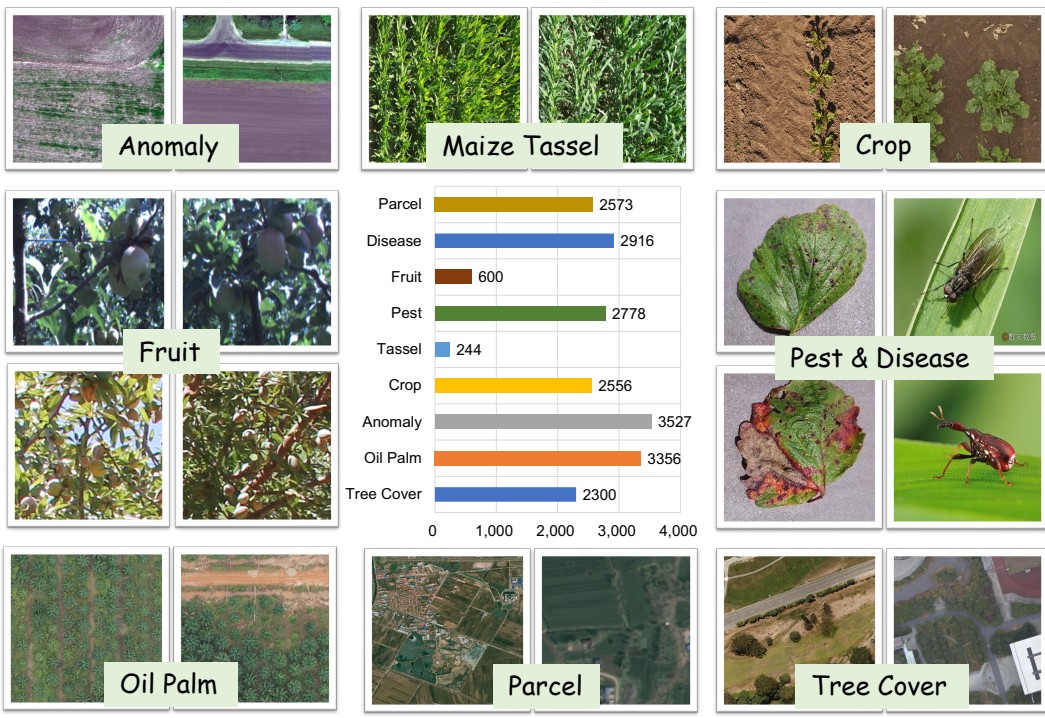

Figure 10: Examples of image datasets, with the bar chart showing the quantity of data in nine detailed subclasses. These subclasses refine the five macro-categories shown in Fig. 3 to microscopically validate task diversity.

vector polygons), this dataset enables a systematic evaluation of agricultural LMMs in multimodal data fusion scenarios.

**Multi-dimensional evaluation framework.** Our benchmark employs a three-dimensional Spatial–Object–Scene framework to establish a hierarchical evaluation system for core agricultural remote-sensing tasks, systematically assessing large-scale multimodal models' (LMMs) capabilities in localization, comprehension, and reasoning within agricultural contexts. In the Spatial dimension, both global canopy distribution from OAM-TCD Dataset[48] and cropland vector data from the Global Parcel Dataset support regional area computation and coverage-rate statistics; the latter's parcel-boundary delineation tasks further challenge LMMs' multi-scale geospatial parsing abilities. In the Object dimension, by leveraging instance-level annotations, the benchmark captures the characteristics of most individual agricultural targets and evaluates models' precision in detection, classification, and counting within agricultural scenarios. Specifically, the Maize Tassel Identification employs YOLO annotations to construct 3×3 grid–based yield-prediction tasks, validating models' dense-object localization and associative inference capabilities. Pest-classification tasks in the IP102 Dataset[52] and 2018 AI Challenge Datasets, together with plant-occlusion analysis tasks that PhenoBench Dataset[51] progressively generates from pixel-level visibility annotations, encompass an object-level evaluation spanning basic attribute discrimination to higher-order relational reasoning. In the Scene dimension, we probe models' capacity for multi-level semantic decomposition, cross-element relational inference, and dynamic decision support in complex agricultural environments. Tasks include multi-scene coverage-rate comparison and distribution inference using OAM-TCD Dataest[48] canopy data, as well as plant-occlusion analysis based on PhenoBench's pixel-level maps and visibility metrics—requirements that ensure robust understanding and reasoning in interactive scenarios, thereby underpinning reliable decision support for precision agriculture.

**Temporal and phenological coverage.** AgroMind establishes a comprehensive temporal axis essential for modeling crop phenodynamics, systematically capturing growth transitions, anomaly evolution, and seasonal patterns through multi-source heterogeneous data. Specifically: (i) Growth-stage representation leverages the OilPalmUAV[49] Dataset to construct QA pairs tracking developmental phases from growth, maturity to aging, while competition management tasks at critical phenophases utilize PhenoBench's visibility metrics[51] to quantify weed-crop interactions; (ii) Anomaly temporal

sensitivity encodes time-critical states through spectral signatures—waterlogging detection via near-infrared absorption characteristics in Agriculture Vision Challenge[50]—enabling cross-temporal implicit comparison; (iii) Multi-seasonal imagery combines the cultivated land status of multiple provinces in China at different time series (7 provincial ground datasets), the Global Ecoregion Tree Monitoring Imagery (OAM-TCD[48]), and the CropHarvest dataset[54] with its annual satellite imaging cycle capturing spectral band evolution of land parcel types, to cover the crop cycle and support phenological transition modeling. This temporal richness enables evaluation of longitudinal reasoning capabilities, particularly for yield prediction and climate adaptation analysis, establishing a foundation for multi-temporal agricultural assessment.

Table 4: Detailed description of datasets used in the AgroMind benchmark

| Dataset | Year | Key Features | Resolution | Link |
|---|---|---|---|---|
| ACFR Orchard Fruit Dataset [53] | 2016 | CSV-format annotations for almond, apple, mango; Fruit radius metadata; Quadrant-based distribution analysis | 308×202, 500×500, etc. | `https://data. acfr.usyd.edu. au/ag/treecrops/ 2016-multifruit/` |
| 2018 AI Challenge Dataset | 2018 | 26 disease types across 9 crops; Disease severity labels (mild, moderate, severe); Deduplicated QA templates for robustness | Variable | `https: //aistudio. baidu.com/ datasetdetail/ 76075` |
| IP102 Dataset [52] | 2019 | 102 insect pest categories; VOC-format annotations; Multi-choice QA templates for pest-crop association | Variable | `https://github. com/xpwu95/IP102` |
| Agriculture Vision Challenge [50] | 2020 | Six anomaly types (e.g., Weed Cluster); Multi-spectral data (RGB + NIR); Binary masks for anomaly detection | 512×512 | `https://github. com/SHI-Labs/ Agriculture-Vision` |
| OilPalmUAV Dataset [49] | 2021 | UAV-captured oil palm trees; Instance-level bbox annotations for 5 growth states (Healthy, Dead, Mismanaged, etc.) | 1024×1024 | `https://github. com/rs-dl/MOPAD` |
| CropHarvest Dataset | 2021 | Global-scale agricultural dataset; Multi-modal remote sensing data (Sentinel-1/2, SRTM, ERA5); Binary and multiclass crop labels; Temporal sequences (12 months) | 896x832, 960x896 | `https: //zenodo.org/ records/5828893` |
| PhenoBench Dataset [51] | 2023 | Hierarchical annotations: semantic segmentation (crop/weed), instance segmentation (plants/leaves), pixel-level visibility maps | 1024×1024 | `https://github. com/PRBonn/ phenobench` |
| OAM-TCD Dataset [48] | 2024 | Global tree cover maps; Semantic segmentation masks; MS-COCO annotations (bbox, polygons); Geospatial metadata (EPSG:3395) | 2048×2048 | `https: //zenodo.org/ records/11617167` |
| Maize Tassel Identification | 2025 | High-resolution maize tassel images; YOLO-format bbox annotations; Cross-image comparison templates for yield prediction | 2736×1824 | `https: //aistudio. baidu.com/ datasetdetail/ 296690` |
| Global Parcel Dataset | 2025 | Multi-spectral GeoTIFF images; .shp parcel polygons; Reprojected to EPSG:3857; Coverage analysis for farmland (After further supplementation, the distribution of the dataset covers not only seven provinces in China, but also some regions in Southeast Asia and Europe.) | Variable | Internal Collection |

## B.2 Data pre-processing details

To address the heterogeneity inherent in multi-source inputs, we propose a unified pre-processing framework characterized by both diversity and depth. The framework consists of four sequential stages—(i) multilevel annotation refinement, (ii) QA schema harmonization, (iii) logical reasoning enhancement, and (iv) task alignment—each contributing to improved data fidelity and downstream model robustness. First, given that each source employs distinct labels, formats, and pre-processing

procedures, we manually refine most original annotations and a subset of images to derive secondary and tertiary annotations. For instance, pre-processing the near-infrared channel data of the Agriculture Vision Challenge[50] collection, the semantic masks from PhenoBench Dataset[51], and the full-band satellite imagery data of the CropHarvest[54] (using GDAL for reading and performing pixel-wise normalization to grayscale representation) facilitates extraction of instance-level annotations for coverage areas and individual objects. Second, QA pair generation is managed via JSON-formatted label files: each file aggregates multiple entries, with each entry linking an image, a question, and its corresponding answer, thereby ensuring high operational efficiency. Crucially, the logical reasoning enhancement stage embeds artificially pre-defining hierarchical reasoning processes into QA design: rather than directly using raw labels as answers, we construct step-by-step reasoning paths (e.g., for canopy diameter calculation: locate bounding box → identify tree → measure pixel coverage → convert to physical scale; for weed control: quantify coverage → analyze occlusion → generate agronomic decisions). Finally, through hierarchical technology integration and task-driven optimization, the pre-processing pipeline transforms raw inputs into a sophisticated, logically coherent evaluation benchmark. This approach not only addresses the challenges of format heterogeneity but also enhances data robustness via targeted augmentation strategies, providing comprehensive support for the robust training and precise evaluation of large multimodal models.

**ACFR Orchard Fruit Dataset.** We ingested CSV-formatted annotations containing fruit radius measurements and computed each fruit's equivalent radius to filter oversized instances for QA pair generation. Concurrently, images were partitioned into four spatial quadrants to derive region-level distribution metrics, facilitating regional distribution inference queries. Finally, we superimposed random noise patches corresponding to various fruit species to introduce distractors in multiple-choice species classification tasks.

**2018 AI Challenge Dataset.** We parsed original leaf-image annotations to extract crop species, disease category, and severity level, after first cleansing augmented samples by excluding flipped, rotated, and duplicated images identified through filename markers. Using spatial disease maps, we instantiated QA templates for crop classification, health-status inference, and severity quantification. To guarantee comprehensive coverage, we applied stratified downsampling to the generated QA pairs—ensuring each crop–disease combination is represented—and then randomly sampled the balanced set to yield a diverse and complete QA corpus.

**IP102 Dataset.** We parsed and indexed VOC-formatted annotations encompassing 102 pest categories. To formulate identification challenges, we sampled distractor species from the same taxonomic superclasses as each ground-truth pest, thereby generating multiple-choice queries probing pest identity and associated crop relevance. Additionally, leveraging the dataset's pest count annotations, we instantiated enumeration queries to assess quantitative reasoning.

**Agriculture Vision Challenge.** We ingested per-image binary masks for six anomaly categories—cloud shadow, double planting, skipped seeding, waterlogging, watercourse, and weed clusters—alongside their multispectral (RGB + NIR) counterparts. For each anomaly type, we extracted mask contours to generate secondary localization annotations and computed metrics quantifying region area and spatial distribution. These statistics, in conjunction with boundary masks data, underpin diversity-driven queries (e.g., anomaly presence detection, coverage estimation). Furthermore, we identified images containing multiple anomalous regions within the same scene, applied positional annotations to assess region overlap against a predefined threshold, and constructed question–answer pairs probing the relative spatial arrangement of anomalous objects.

**OilPalmUAV Dataset.** We parsed the original annotation's bounding-box coordinates and employed PIL's ImageDraw module to overlay red bounding boxes scaled by a factor of 1.1–1.5, thereby precluding trivial "box-based" cues. Next, using these augmented boxes, we formulated growth-stage classification tasks for each target tree. Finally, we enumerated all target instances per image to derive scene-level tree counts.

**CropHarvest Dataset.** We ingested GEOJSON-formatted annotations containing binary and multiclass crop labels, along with spatiotemporal metadata. To enable visual question answering, we processed multi-spectral time-series imagery by decomposing each GeoTIFF into constituent spectral bands and generating normalized band mosaics through four critical steps: (i) parsing Sentinel-1/2 data using GDAL; (ii) applying per-band min-max normalization to mitigate sensor variation; (iii) constructing band mosaics while filtering degenerate cases (e.g., uniform-value bands); (iv) validating non-zero pixel ratios to exclude invalid observations— ensuring maximal geographic information

retention for large-scale models. We subsequently leveraged polygon geometries and crop presence labels (is_crop) to instantiate two complementary QA modalities: binary classification via balanced queries (50% crop and 50% non-crop QA pairs) verifying agricultural land use, and spatial quantification through polygon-counting tasks assessing field fragmentation .

Table 5: Models grouped by family.

| Model Family | Model Name | Year | Parameters | Link |
|---|---|---|---|---|
| **Close-sourced, API** | | | | |
| GPT-4 | GPT-4o | 2023 | N/A | `https://platform.openai.com/docs/models/gpt-4o` |
| Gemini | Gemini-1.5-Flash | 2024 | N/A | `https://ai.google.dev/gemini-api/docs/models/gemini` |
| | Gemini-1.5-Pro | 2024 | N/A | `https://ai.google.dev/gemini-api/docs/models/gemini` |
| Claude | Claude-3.5-Sonnet | 2024 | N/A | `https://docs.anthropic.com/en/docs/about-claude/models/all-models` |
| **Open-sourced** | | | | |
| DeepSeek-VL2 | DeepSeek-VL2-tiny | 2024 | 1B | `https://huggingface.co/deepseek-ai/deepseek-vl2-tiny` |
| | DeepSeek-VL2-small | 2024 | 2.8B | `https://huggingface.co/deepseek-ai/deepseek-vl2-small` |
| TinyLLaVA | TinyLLaVA | 2024 | 3.1B | `https://huggingface.co/tinyllava/TinyLLaVA-Phi-2-SigLIP-3.1B` |
| InternVL2 | InternVL2-2B | 2024 | 2B | `https://huggingface.co/OpenGVLab/InternVL2-2B` |
| | InternVL2-4B | 2024 | 4B | `https://huggingface.co/OpenGVLab/InternVL2-4B` |
| | InternVL2-8B | 2024 | 8B | `https://huggingface.co/OpenGVLab/InternVL2-8B` |
| | InternVL2-26B | 2024 | 26B | `https://huggingface.co/OpenGVLab/InternVL2-26B` |
| XComposer2 | XComposer2-4KHD | 2023 | 7B | `https://huggingface.co/internlm/internlm-xcomposer2-4khd-7b` |
| LLaVA-NeXT | LLaVA-NeXT-7B-Mistral | 2024 | 7B | `https://huggingface.co/llava-hf/llava-v1.6-mistral-7b-hf` |
| | LLaVA-NeXT-7B-Vicuna | 2024 | 7B | `https://huggingface.co/llava-hf/llava-v1.6-vicuna-7b-hf` |
| | LLaVA-NeXT-8B | 2024 | 8B | `https://huggingface.co/llava-hf/llama3-llava-next-8b-hf` |
| | LLaVA-NeXT-13B | 2024 | 13B | `https://huggingface.co/llava-hf/llava-v1.6-vicuna-13b-hf` |
| | LLaVA-NeXT-34B | 2024 | 34B | `https://huggingface.co/llava-hf/llava-v1.6-34b-hf` |
| InstructBLIP | InstructBLIP-Vicuna-7B | 2023 | 7B | `https://huggingface.co/Salesforce/instructblip-vicuna-7b` |
| LLaVA-NeXT-Interleave | LLaVA-NeXT-Interleave-7B | 2024 | 7B | `https://huggingface.co/llava-hf/llava-interleave-qwen-7b-hf` |
| Mantis | Mantis-LLaMA3-SigLIP | 2024 | 8B | `https://huggingface.co/TIGER-Lab/Mantis-8B-siglip-llama3` |
| | Mantis-Idefics2 | 2024 | 8B | `https://huggingface.co/TIGER-Lab/Mantis-8B-Idefics2` |
| Idefics2 | Idefics-2-8b | 2024 | 8B | `https://huggingface.co/HuggingFaceM4/idefics2-8b` |
| GeoChat | GeoChat | 2024 | 7B | `https://huggingface.co/MBZUAI/geochat-7B` |
| GeoLLaVA-8K | GeoLLaVA-8K | 2025 | 7B | `https://huggingface.co/initiacms/GeoLLaVA-8K` |

**PhenoBench Dataset.** In this dataset, Semantic segmentation masks were parsed to differentiate crop, weed, partially visible crop and partially visible weed regions. Instance segmentation annotations were aggregated to enumerate individual plants and leaves. Plant completeness and inter-plant

occlusion were quantified via pixel-level visibility maps, and responses were validated against predefined visibility thresholds. This workflow yields a multidimensional QA dataset encompassing plant counting, occlusion assessment, and weed symbiosis evaluation.

**OAM-TCD Dataset.** In the original dataset, we selected a training subset of approximately 2,000 images for segmentation and converted the source GeoTIFF tree cover files to JPEG format using the Pillow library. All images were uniformly cast to RGB color space and saved at 95% quality to balance storage requirements and visual fidelity. To derive foundational tasks—namely, ecoregion classification and individual plant identification—we employed the dataset's initial annotations. Additionally, by leveraging semantic segmentation outputs alongside MS COCO–style polygon annotations, we divided each image into a 3 × 3 spatial grid and computed region-level statistics, specifically per-cell and overall canopy cover. These statistics serve as the basis for spatial distribution inference tasks. Finally, for each image, we randomly sampled four grid cells to generate multi-image visual comparison queries.

**Maize Tassel Identification.** In this dataset, YOLO-style bounding boxes were parsed to localize tassel centroids and extract bounding dimensions. Then, each image was partitioned into a 3 × 3 spatial grid to derive tassel distribution metrics. Next, threshold-driven heuristics (e.g., "tassel count > 10 → potential yield reduction") instantiated yield prediction queries. Finally, for cross-image visual comparisons, two grid cells per image were randomly sampled to construct question–answer pairs.

**Global Parcel Dataset.** We reprojected all GeoTIFF multispectral images and associated shapefile polygons to the EPSG:3857 coordinate system for consistent spatial alignment. For cropland parcels within each study area, we generated random rectangular query regions with sizes ranging from 10% to 20% of the area's minimum side length, ensuring strict non-overlapping using a buffer-based deduplication mechanism. These regions were exported as PNGs and used to compute land-cover metrics—parcels count, total cropland area, and coverage—to formulate cropland enumeration, area estimation, and related tasks. In a secondary selection phase, we manually curated high-fidelity cropland maps with their xmin and ymin anchored at the upper-left corner, and defined parcel-boundary delineation queries by normalizing bounding boxes according to original cropland distribution labels.

# C    Details of AgroMind benchmark

## C.1    Human participants

We randomly divided our dataset into ten equal and non-overlapping subsets. Each subset was answered by at least three different volunteers. We did not provide any manual instructions apart from specifying the required answer format, such as: "Please select one correct answer from the options below." All volunteers participated on a voluntary basis, and we provided compensation that exceeded the minimum standard.

## C.2    Evaluation models and computational resource

We compared different models across multiple tasks in our benchmark to understand their capabilities in agricultural scenarios. We evaluated open-source models, including DeepSeek-VL2 series[62], TinyLLaVA[56], InternVL2 series[30], XComposer2[57], LLaVA-NeXT series[32], InstrucBLIP[58], LLaVA-NeXT-Interleave[31], Mantis series[63], Idefics2[59], GeoChat [60], and GeoLLaVA-8K, as well as closed-source models such as GPT-4o[11], Gemini[12], and Claude[55]. All experiments were conducted using an NVIDIA A800 GPU. Table 5 provides details of these models, and the specific computational resource requirements for each model are included in their respective official URLs listed in the table.

## C.3    Definitions of question difficulty levels

To facilitate a more detailed analysis of model performance, we define three levels of question difficulty based on human accuracy. A question is considered easy if fewer than one-third of the volunteers answered it incorrectly, medium if between one-third and two-thirds of the volunteers answered it incorrectly, and hard if more than two-thirds of the volunteers answered it incorrectly.

Table 6: Performance comparison with aggregated question types.

| Model | Judgement | Counting | Text Choice | Image Choice | Open-ended | Overall |
|---|---|---|---|---|---|---|
| Human | 53.52 | 23.35 | 38.97 | 40.80 | 5.80 | 33.15 |
| Random | 51.98 | 1.32 | 20.79 | 24.04 | 0.00 | 19.24 |
| GPT-4o [11] | **69.80** | 23.77 | **51.48** | **41.80** | 15.14 | **43.14** |
| Gemini-1.5-Flash [12] | 65.35 | 23.96 | 48.45 | 38.25 | 16.33 | 40.92 |
| Gemini-1.5-Pro [12] | 65.84 | 22.64 | 49.53 | 38.80 | 13.15 | 41.06 |
| Claude-3.5-Sonnet [55] | 65.84 | 25.28 | 41.59 | 32.79 | 18.73 | 37.11 |
| DeepSeek-VL2-tiny [62] | 64.36 | 20.94 | 28.40 | 24.59 | 10.76 | 27.51 |
| DeepSeek-VL2-small [62] | 65.84 | 24.72 | 28.80 | 23.22 | 7.57 | 28.08 |
| TinyLLaVA [56] | 60.40 | 19.43 | 32.17 | 22.13 | 3.98 | 28.01 |
| InternVL2-2B [30] | 61.88 | 22.08 | 30.96 | 21.31 | 9.96 | 28.40 |
| InternVL2-4B [30] | 61.88 | 21.32 | 36.27 | 23.50 | 7.97 | 31.15 |
| XComposer2-4KHD [57] | 57.43 | 20.00 | 37.21 | 30.87 | 19.12 | 33.02 |
| LLaVA-NeXT-7B-Mistral [32] | 57.92 | 20.38 | 33.11 | 23.50 | 9.56 | 29.17 |
| LLaVA-NeXT-7B-Vicuna [32] | 55.94 | 23.21 | 24.23 | 22.13 | **21.91** | 25.82 |
| InstructBLIP-Vicuna-7B [58] | 37.13 | 6.98 | 18.24 | 24.59 | 7.97 | 17.39 |
| LLaVA-NeXT-Interleave-7B [31] | 49.01 | 11.32 | 17.50 | 22.40 | 15.14 | 19.01 |
| Mantis-LLaMA3-SigLIP [63] | 56.93 | 19.62 | 37.35 | 26.78 | 4.78 | 31.18 |
| Mantis-Idefics2 [63] | 53.96 | 16.23 | 36.68 | 24.59 | 12.75 | 30.41 |
| LLaVA-NeXT-8B [32] | 59.90 | 22.64 | 36.47 | 22.68 | 11.16 | 31.53 |
| InternVL2-8B [30] | 60.89 | 24.15 | 43.88 | 31.69 | 3.98 | 36.30 |
| Idefics-2-8b [59] | 66.34 | 22.08 | 28.47 | 21.86 | 5.58 | 27.09 |
| LLaVA-NeXT-13B [32] | 64.85 | 20.57 | 34.99 | 23.22 | 9.96 | 30.69 |
| InternVL2-26B [30] | 68.32 | **27.55** | 43.20 | 30.87 | 6.77 | 37.25 |
| LLaVA-NeXT-34B [32] | 57.43 | 26.04 | 42.66 | 24.04 | 12.35 | 35.52 |
| GeoChat [60] | 57.92 | 19.81 | 28.87 | 21.86 | 1.59 | 25.93 |
| GeoLLaVA-8K [61] | 53.47 | 0.00 | 17.09 | 22.40 | 0.80 | 15.73 |

## C.4   More details of evaluation protocol

**Single-choice questions.** For single-choice questions, the predicted answer is identified by locating the chosen option (e.g., A, B, C, D) in the model output. If option markers such as '(A)' or 'A:' are present, they are extracted directly too. Otherwise, the system searches for the appearance of option content in the text, selecting the last mentioned one if multiple are present. The predicted option is compared to the ground-truth label, and the answer is marked correct if they match exactly.

**Multiple-choice questions.** For multiple-choice questions, the evaluation logic allows for multiple correct options. All identifiable choices mentioned in the output are extracted and compared as a set to the reference set. A prediction is considered correct only if the extracted set of options matches the ground truth exactly, with no missing or extra items.

**Short-answer and semi-open questions.** For short-answer and semi-open questions, a keyword matching approach is employed. The reference and predicted answers are segmented into clauses, and key elements such as named entities or numerical values are extracted. Before matching, all text is normalized by removing punctuation, standardizing numerical formats, and converting to lowercase. A prediction is considered correct if all key elements in the reference answer can be matched to semantically equivalent expressions in the model output.

**Open-ended questions.** For open-ended questions, we have clear and fixed reference answers. We consider a prediction correct only when it is semantically consistent with the reference answer. Therefore, we use the paraphrase-MiniLM-L6-V2 [64] model to compute cosine similarity between predicted and reference answers. We consider an answer correct only when similarity exceeds the set threshold. Through preliminary experiments on a dataset subset, we determined the threshold as 0.60, under which all models achieve reasonable accuracy. We plan to employ expert human evaluation and calculate the overlap rate between expert and model judgments to determine the optimal threshold that maximizes agreement between expert and model.

**Visual localization questions.** For questions requiring spatial localization, the predicted bounding box coordinates are extracted from the output and compared against the reference using the Inter-

Table 7: Performance comparison with aggregated question scenes.

| Model | Parcel | Disease | Fruit | Pest | Tassel | Crop | Anomaly | Oil Palm | Tree Cover | Overall |
|---|---|---|---|---|---|---|---|---|---|---|
| Human | 15.83 | 28.96 | 42.84 | 54.53 | 18.83 | 12.22 | 28.48 | **42.17** | 45.93 | 33.15 |
| Random | 16.57 | 25.65 | 12.20 | 26.72 | 16.74 | 19.42 | 17.56 | 12.33 | 26.72 | 19.24 |
| GPT-4o [11] | 26.47 | 57.65 | 50.79 | 73.28 | 26.09 | 31.13 | **45.43** | 18.00 | 61.45 | **43.14** |
| Gemini-1.5-Flash [12] | 18.87 | **59.07** | 51.18 | 75.29 | 22.53 | 27.15 | 37.94 | 22.67 | 59.54 | 40.92 |
| Gemini-1.5-Pro [12] | 27.45 | 50.18 | 48.03 | **77.30** | 22.53 | 31.46 | 33.72 | 20.33 | **62.21** | 41.06 |
| Claude-3.5-Sonnet [55] | 25.74 | 44.48 | 38.19 | 64.37 | **32.02** | 30.13 | 24.59 | 26.67 | 54.96 | 37.11 |
| DeepSeek-VL2-tiny [62] | 15.20 | 20.64 | 40.55 | 57.76 | 28.46 | 18.87 | 10.07 | 31.67 | 33.97 | 27.51 |
| DeepSeek-VL2-small [62] | 15.44 | 25.27 | 41.73 | 60.92 | 27.27 | 8.61 | 18.03 | 27.33 | 34.35 | 28.08 |
| TinyLLaVA [56] | 20.83 | 19.57 | 47.64 | 60.63 | 12.25 | 13.58 | 28.81 | 10.00 | 37.02 | 28.01 |
| InternVL2-2B [30] | 16.91 | 26.33 | 42.13 | 45.11 | 23.72 | 16.56 | 35.36 | 17.33 | 32.44 | 28.40 |
| InternVL2-4B [30] | 21.32 | 36.65 | 38.19 | 51.15 | 21.74 | 15.23 | 35.60 | 14.33 | 46.56 | 31.15 |
| XComposer2-4KHD [57] | 27.45 | 33.45 | 40.55 | 54.60 | 22.13 | **35.10** | 26.23 | 20.33 | 38.93 | 33.02 |
| LLaVA-NeXT-7B-Mistral [32] | 25.49 | 32.74 | 38.19 | 43.68 | 14.62 | 13.58 | 35.13 | 16.33 | 40.08 | 29.17 |
| LLaVA-NeXT-7B-Vicuna [32] | 20.83 | 24.91 | 30.31 | 50.86 | 26.09 | 20.86 | 16.39 | 11.33 | 34.35 | 25.82 |
| InstructBLIP-Vicuna-7B [58] | 14.71 | 17.79 | 22.05 | 31.61 | 10.67 | 10.26 | 15.46 | 10.67 | 23.28 | 17.39 |
| LLaVA-NeXT-Interleave-7B [31] | 17.65 | 18.15 | 19.69 | 29.89 | 20.16 | 23.18 | 9.60 | 8.00 | 29.01 | 19.01 |
| Mantis-LLaMA3-SigLIP [63] | 23.77 | 27.40 | 44.09 | 63.22 | 24.90 | 30.79 | 18.74 | 12.00 | 40.46 | 31.18 |
| Mantis-Idefics2 [63] | 21.81 | 38.79 | 44.49 | 61.78 | 13.44 | 24.50 | 13.82 | 20.67 | 40.84 | 30.41 |
| LLaVA-NeXT-8B [32] | **29.41** | 23.49 | **54.33** | 52.59 | 18.58 | 29.14 | 27.17 | 12.67 | 37.40 | 31.53 |
| InternVL2-8B [30] | 19.61 | 35.94 | 44.49 | 67.82 | 23.72 | 28.15 | 37.24 | 16.33 | 55.73 | 36.30 |
| Idefics-2-8b [59] | 18.63 | 28.11 | 44.88 | 58.05 | 18.18 | 9.93 | 13.35 | 20.00 | 39.69 | 27.09 |
| LLaVA-NeXT-13B [32] | 24.51 | 27.40 | 44.49 | 51.15 | 17.79 | 22.52 | 35.36 | 8.33 | 43.13 | 30.69 |
| InternVL2-26B [30] | 21.32 | 38.43 | 41.34 | 54.89 | 29.64 | 27.15 | 38.17 | 34.33 | 54.20 | 37.25 |
| LLaVA-NeXT-34B [32] | **29.41** | 35.23 | 47.64 | 58.91 | 20.95 | 28.15 | 36.53 | 13.00 | 49.24 | 35.52 |
| GeoChat [60] | 18.14 | 28.47 | 37.40 | 50.00 | 17.00 | 21.52 | 15.93 | 11.00 | 39.31 | 25.93 |
| GeoLLaVA-8K [61] | 13.24 | 13.17 | 11.02 | 26.72 | 14.62 | 8.28 | 14.29 | 15.00 | 25.19 | 15.73 |

section over Union (IoU) metric. A prediction is marked correct if the IoU exceeds 0.5, indicating sufficient overlap between the predicted and ground-truth regions.

## C.5 More experimental results and discussions

Table 6 compares the performance of humans, random guesses, and LMMs across different types of questions. The accuracy rates of the random guesses show that our dataset ensures a uniform distribution of answers. For instance, it achieves an accuracy of about 50% on judgement type questions, as these only have two options: yes or no. For image choice questions, the accuracy is approximately 25% because most such questions have four options. The accuracy for text choice is slightly lower because some questions have more than four options. However, GPT-4o consistently achieves the best performance in judgement, text choice, and image choice, which demonstrates its superior ability to understand both textual and visual content compared to other LMMs. Notably, DeepSeek-VL2-small and DeepSeek-VL2-tiny perform poorly overall but achieve remarkable performance on judgement type questions. This indicates that current LMMs have significant specialization issues and cannot fully replace humans in the agricultural scenery.

Table 7 shows the performance of humans and various LMMs across different scenes. Based on the content of the questions and images, we divide our dataset into distinct scenes. It becomes clear that different models excel in different scenes, as the highest accuracy in each scene is distributed across various LMMs. For example, Xcomposer2-4KHD achieves the highest performance in Crop-related scenarios, Claude-3.5-Sonnet performs best in Tassel, and Gemini-1.5-Pro leads in both Pest and Tree Cover scenes. However, GPT-4o maintains a dominant lead, achieving the highest accuracy in Anomaly scene and overall performance. This table further shows that our dataset covers a wide range of agricultural scenes which is a pattern that is very common in real-world applications, making it challenging for LMMs to perform well across all of them.

Table 8 illustrates the performance of different models on images captured by various sensors including unmanned aerial vehicles (UAV), satellites, and cameras. On UAV-captured images, where the objects are typically smaller, and the context tends to be more complex due to higher variability in altitude, angles, and resolutions, most models struggle significantly. In contrast, human participants demonstrate a notably higher understanding of these images, achieving higher performance than most models. However, for SL and CM images, which generally have clearer object representations and more consistent viewpoints, the performance of human evaluators is markedly lower than that of LMMs. This discrepancy indicates that LMMs excel at interpreting more structured and standardized

Table 8: Performance comparison with aggregated question sensors.

| Model | UAV | Satellites | Camera | Overall |
|---|---|---|---|---|
| Human | 26.00 | 28.89 | 41.22 | 33.15 |
| Random | 14.73 | 20.24 | 21.05 | 19.24 |
| GPT-4o [11] | 25.47 | **40.94** | 59.24 | **43.14** |
| Gemini-1.5-Flash [12] | 26.34 | 34.21 | **60.65** | 40.92 |
| Gemini-1.5-Pro [12] | 25.76 | 37.16 | 57.62 | 41.06 |
| Claude-3.5-Sonnet [55] | 30.97 | 31.01 | 49.73 | 37.11 |
| DeepSeek-VL2-tiny [62] | 27.35 | 18.05 | 40.11 | 27.51 |
| DeepSeek-VL2-small [62] | 20.55 | 21.00 | 43.03 | 28.08 |
| TinyLLaVA [56] | 10.71 | 27.24 | 41.95 | 28.01 |
| InternVL2-2B [30] | 18.09 | 27.32 | 37.51 | 28.40 |
| InternVL2-4B [30] | 16.35 | 31.50 | 41.73 | 31.15 |
| XComposer2-4KHD [57] | 28.08 | 28.88 | 42.16 | 33.02 |
| LLaVA-NeXT-7B-Mistral [32] | 14.04 | 31.42 | 37.51 | 29.17 |
| LLaVA-NeXT-7B-Vicuna [32] | 18.81 | 22.15 | 35.89 | 25.82 |
| InstructBLIP-Vicuna-7B [58] | 11.29 | 16.16 | 23.57 | 17.39 |
| LLaVA-NeXT-Interleave-7B [31] | 16.06 | 17.56 | 23.14 | 19.01 |
| Mantis-LLaMA3-SigLIP [63] | 22.72 | 25.92 | 44.43 | 31.18 |
| Mantis-Idefics2 [63] | 21.27 | 22.64 | 47.46 | 30.41 |
| LLaVA-NeXT-8B [32] | 20.12 | 29.61 | 42.59 | 31.53 |
| InternVL2-8B [30] | 23.30 | 33.96 | 49.08 | 36.30 |
| Idefics-2-8b [59] | 15.63 | 21.58 | 42.92 | 27.09 |
| LLaVA-NeXT-13B [32] | 15.92 | 32.08 | 39.89 | 30.69 |
| InternVL2-26B [30] | **33.00** | 34.13 | 44.54 | 37.25 |
| LLaVA-NeXT-34B [32] | 20.41 | 35.44 | 46.92 | 35.52 |
| GeoChat [60] | 16.50 | 22.23 | 37.84 | 25.93 |
| GeoLLaVA-8K [61] | 11.87 | 16.90 | 17.08 | 15.73 |

image inputs, whereas their performance sharply declines with UAV imagery characterized by greater complexity and variability. Therefore, further enhancements in understanding UAV-captured images remain a critical area for future research and development.

The observed performance imbalance across agricultural tasks has significant implications for fairness and potential biases in deploying LMMs for global agricultural applications. Given AgroMind's extensive geographic and agricultural diversity, performance disparities in tasks like spatial localization and quantitative reasoning may disproportionately affect regions relying heavily on precision agriculture and resource optimization. Specifically, regions characterized by complex agricultural landscapes, smaller-scale farming systems, or diverse polycultures—often prevalent in parts of Africa, Southeast Asia, and Latin America—might not fully benefit from current model capabilities, exacerbating existing technological divides. Therefore, it is critical to address these imbalances through targeted model training and fine-tuning, ensuring equitable performance across different agricultural contexts. Enhancing the robustness and adaptability of LMMs in challenging scenarios, such as UAV-captured imagery, will be essential to prevent bias and promote fair access to advanced agricultural technologies worldwide.

## D   Limitations and future work

While AgroMind provides a comprehensive benchmark for evaluating the perception and reasoning capabilities of LMMs, there are several limitations to consider. Compared to the diverse sensor requirements in real-world agriculture, AgroMind are limited to RGB&NIR satellite images, UAV photography, and ground cameras. Key technologies such as hyperspectral images or SAR (Synthetic Aperture Radar) are absent, potentially reducing its capacity to replicate the spectral and spatial diversity of actual field conditions. Additionally, the English-centric datasets and question generation restrict accessibility for non-English-speaking users. Future work could enrich AgroMind with multi-sensor data and multilingual support. Furthermore, we aim to develop domain-adapted agricultural LMMs, to promote the reasoning ability in real-world agricultural scenarios.

## E   Case study

In this section, we provide some question cases of different AgroMind tasks and the responses from GPT-4o and InternVL2-26B **(see Fig. 11 - Fig. 36).**

GPT-4o is the best-performing closed-source model evaluated on AgroMind, and it also represents the current SOTA in general-purpose multimodal understanding. InternVL2-26B is the strongest open-source model in our evaluation, achieving the highest overall accuracy among open-access LMMs. By comparing these two models, we aim to provide insights into both the upper performance bound of current LMMs and the progress of open-source alternatives. The cases cover various task dimensions from AgroMind. Each case consists of a visual input (satellite or aerial imagery), an associated question designed according to the task type, and the model-generated answers.

Through these examples, we observe that GPT-4o tends to produce more precise and context-aware responses, especially in tasks requiring complex reasoning or spatial interpretation. InternVL2-26B, while competent in visual recognition and straightforward classification tasks, occasionally struggles with nuanced spatial relationships and logical inference. These differences reflect the varying levels of generalization and domain adaptation in current LMMs.

---

📍 Spatial Localization

**Question**: Dividing the image into nine-grids, in which nine-grid region are the trees primarily located based on the area covered by the trees in the image?

**Option**:

(A) Upper left region     (B) Upper center region     (C) Upper right region
(D) Middle left region     (E) Central region     (F) Middle right region
(G) Lower left region     (H) Lower center region     (I) Lower right region

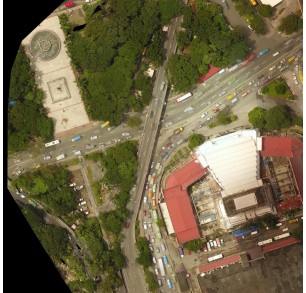

**Answer**: B

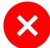

**GPT-4o**: The vast majority of the tree canopy sits in the top-left cell of the 3×3 grid. So the answer is **A: Upper left region.**

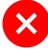

**InternVL2-26B**: Based on the area covered by the trees, **the upper left region (A)** has the most extensive coverage of trees.

Figure 11: A question case of the spatial localization task with responses from GPT-4o and InternVL2-26B.

## 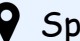 Spatial Localization

**Question**: Knowing that the image has 512x512 pixels, what is the type of anomaly shown at coordinates (450,34)?
**Option**:
(A) cloud_shadow  (B) double_plant  (C) planter_skip  (D) standing_water
(E) Waterway       (F) weed_cluster  (G) No anomaly

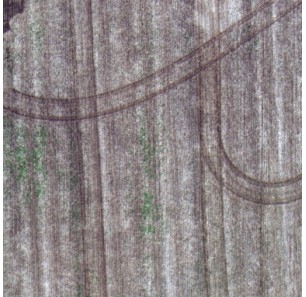

Answer: G

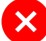

**GPT-4o**: Based on the image and the coordinates (450, 34), the anomaly visible is likely a weed cluster, as there seems to be a pattern of irregular vegetation growth in that area. The most fitting option would be:
 **F: weed_cluster.**

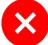

**InternVL2-26B**: The type of anomaly shown at coordinates (450,34) is **(C) planter_skip**.

Figure 12: A question case of the spatial localization task with responses from GPT-4o and InternVL2-26B.

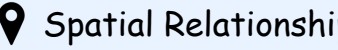

## Spatial Relationship

**Question**: Based on the provided image, if the abnormal area marked as standing_water is taken as the center, what is the spatial relationship between the abnormal area marked as weed_cluster and this central area? (The overlap ratio threshold is 50%.)

**Option**:
| | | | |
|---|---|---|---|
| (A) Above | (B) Below | (C) Left | (D) Right |
| (E) Upper Left | (F) Lower Left | (G) Upper Right | (H) Lower Right |
| (I) Overlap | | | |

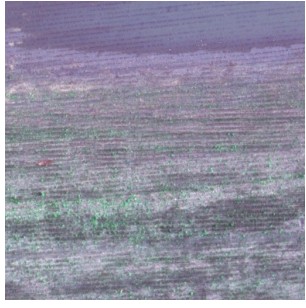

Answer: B

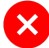

**GPT-4o**: Based on the visual analysis of the image, the area marked as **weed_cluster** appears to be located to the **lower right** of the area marked as **standing_water**. Thus, the most appropriate answer is:
**H: Lower Right.**

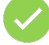

**InternVL2-26B**: **(B) Below**
The weed_cluster is located below the standing_water area in the image.

Figure 13: A question case of the spatial relationship task with responses from GPT-4o and InternVL2-26B.

## 📍 Spatial Relationship

**Question**: Based on the provided image, if the abnormal area marked as waterway is taken as the center, what is the spatial relationship between the abnormal area marked as standing water and this central area? (The overlap ratio threshold is 50%.)

**Option**:
(A) Above         (B) Below         (C) Left         (D) Right
(E) Upper Left    (F) Lower Left    (G) Upper Right    (H) Lower Right
(I) Overlap

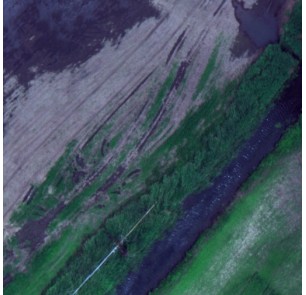

Answer: E

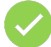

**GPT-4o**: The standing-water patch lies mostly to the upper-left of the long waterway feature (well over 50% of its area falls in that quadrant), so the correct answer is **E: Upper Left.**

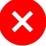

**InternVL2-26B**:  **(A) Above**
The standing water area is located above the central waterway area in the image.

Figure 14: A question case of the spatial relationship task with responses from GPT-4o and InternVL2-26B.

## 📍 Boundary Detection

**Question**: Find the most appropriate rectangular angular coordinates that can enclose all the cultivated land in the image. (The upper left corner is xmin, ymin. The answer format is [xmin, ymin, xmax, ymax].)
**Option**:
(A) [0.334, 0.146, 0.479, 0.681]
(B) [0.085, 0.306, 0.257, 0.656]
(C) [0.257, 0.192, 0.997, 0.902]
(D) [0.0, 0.0, 1.0, 0.446]

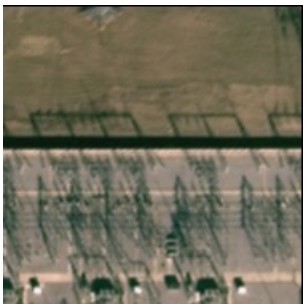

Answer: D

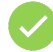

**GPT-4o**: The only box that cleanly covers the bare-soil "cultivated" plots at the top of the frame (and stops just above the dark road/vegetation band) is [0.0, 0.0, 1.0, 0.446], so **the correct choice is D.**

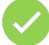

**InternVL2-26B**: **[0.0, 0.0, 1.0, 0.446]**
Explanation: The cultivated land is located in the upper part of the image. The coordinates [0.0, 0.0, 1.0, 0.446] cover the entire upper section, ensuring all cultivated land is enclosed.

Figure 15: A question case of the boundary detection task with responses from GPT-4o and InternVL2-26B.

## 📍 Boundary Detection

**Question**: Find the most appropriate rectangular angular coordinates that can enclose all the cultivated land in the image. (The upper left corner is xmin, ymin. The answer format is [xmin, ymin, xmax, ymax].)

**Option**:
(A) [0.313, 0.589, 0.879, 0.991]
(B) [0.038, 0.541, 0.963, 0.876]
(C) [0.074, 0.082, 0.91, 0.165]
(D) [0.3419, 0.0, 1.0, 1.0]

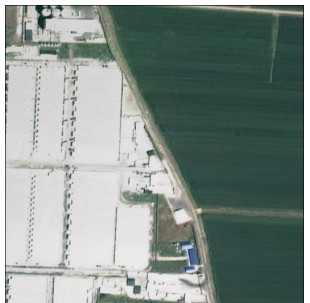

Answer: D

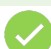

**GPT-4o**: The only box that fully contains the green open-field area (the cultivated land) from top to bottom on the right side is
[0.3419, 0.0, 1.0, 1.0], **so the correct choice is D.**

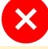

**InternVL2-26B**: **[0.038, 0.541, 0.963, 0.876]**
Explanation: The coordinates [0.038, 0.541, 0.963, 0.876] enclose the entire cultivated land area on the right side of the image, ensuring all the farmland is within the specified rectangle.

Figure 16: A question case of the boundary detection task with responses from GPT-4o and InternVL2-26B.

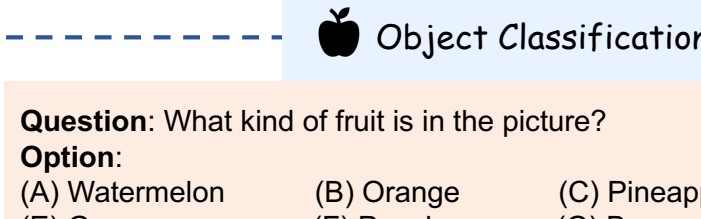

**Question**: What kind of fruit is in the picture?
**Option**:
(A) Watermelon      (B) Orange      (C) Pineapple      (D) Pear
(E) Grape      (F) Peach      (G) Banana      (H) Apple
(I) Overlap

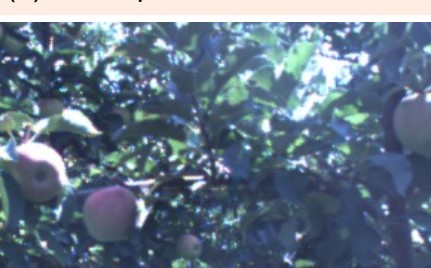

Answer: H

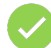

**GPT-4o**: H. The image depicts several round fruits hanging from tree branches, with a smooth skin and a pinkish-red to yellowish-green color gradient, which are typical characteristics of apples. The fruits' size, shape, and the dense, leafy environment also resemble a typical apple orchard setting. These visual clues strongly indicate that the fruits are apples rather than other options like peaches, pears, or grapes.

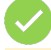

**InternVL2-26B**: **(H): Apple**
The fruits in the picture are apples, identifiable by their round shape and the visible coloration typical of apples.

Figure 17: A question case of the object classification task with responses from GPT-4o and InternVL2-26B.

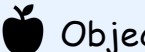 Object Classification

**Question**: Are the pests flax budworm?

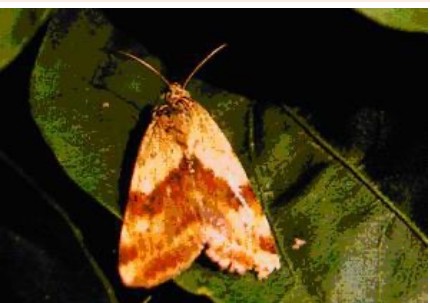

Answer: Yes

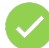

**GPT-4o**:**Yes**. The pest in the image closely resembles the flax budworm (Heliothis ononis), characterized by its triangular-shaped wings, earthy brown and cream coloration, and distinct banding pattern across the wings. These traits match documented appearances of flax budworm moths, which are known agricultural pests affecting flax and related crops. The posture on a leaf and the overall body morphology further support this identification.

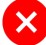

**InternVL2-26B:No**, this is not a flax budworm. The image shows a moth, likely a species of Noctuidae, which are commonly known as owlet moths. Flax budworms are typically caterpillars, not moths.

Figure 18: A question case of the object classification task with responses from GPT-4o and InternVL2-26B.

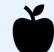 **Pest/Disease Diagnostics**

**Question**: Is the leaf healthy? If not, what disease does it have?
**Option**:
(A) No, it has Frog-eye Spot
(B) No, it has Spotting Leaf Spot Fungus
(C) No, it has Cedar Apple Rust
(D) No, it has Scab

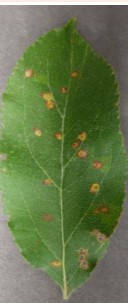

Answer: C

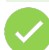

**GPT-4o**:**C**. The leaf shows symptoms that are consistent with Cedar Apple Rust, such as yellowish-orange spots with distinct edges and a web-like appearance. This disease is caused by the fungus Gymnosporangium juniperi-virginianae and typically affects plants in the Rosaceae family, including apples and crabapples. The characteristic orange spots on the leaf are a strong indicator of this condition.

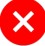

**InternVL2-26B**: **(A)**: No, it has Frog-eye Spot
The leaf shows symptoms of Frog-eye Spot, characterized by small, circular spots with a distinct border.

Figure 19: A question case of the pest/disease diagnostics task with responses from GPT-4o and InternVL2-26B.

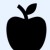 **Pest/Disease Diagnostics**

**Question**: What is the health status of the leaves in the picture?
**Option**:
(A) Healthy
(B) It's diseased, but the status is almost negligible
(C) General
(D) Serious

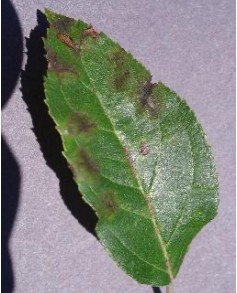

Answer: D

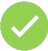

**GPT-4o**: Answer: **D**: serious.The presence of numerous dark spots indicates a more serious disease affecting the leaf.

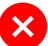

**InternVL2-26B: (B)**: It's diseased, but the status is almost negligible
The leaf shows some signs of disease or damage, such as spots and discoloration, but these issues are minor and do not significantly affect the overall health of the leaf.

Figure 20: A question case of the pest/disease diagnostics task with responses from GPT-4o and InternVL2-26B.

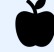 Growth Status Recognition

**Question**: How many large almonds with a radius greater than 15.0 are there?

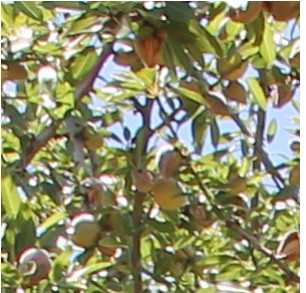

Answer: 6

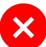

**GPT-4o**: **0**. The image shows a tree with several almonds hanging from its branches. Upon close inspection, none of the visible almonds appear to have a radius greater than 15.0 (the size and focus suggest they are smaller). Since the image doesn't provide a clear indication of any almonds that meet the size requirement, the count is 0.

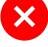

**InternVL2-26B:** There are **no large almonds** with a radius greater than 15.0 in the image. The objects appear to be smaller nuts or seeds.

Figure 21: A question case of the growth status recognition task with responses from GPT-4o and InternVL2-26B.

**Question**: What is the status of the tree in the red box in the image?
**Option**:
(A) Dead
(B) Healthy
(C) Grass,which is crowded with grass
(D) Small,which crown diameter is less than 6.4m
(E) Yellow

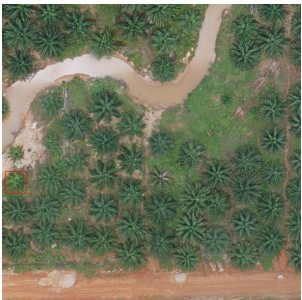

Answer: E

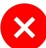

**GPT-4o**: **B**. The tree in the red box shows a full, symmetrical crown with dark green, healthy foliage. Its crown size is comparable to neighboring mature trees, and there are no visible signs of yellowing, dieback, or overgrowth by grass. The consistent density and shape of the canopy confirm that the tree is healthy.

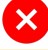

**InternVL2-26B: (A): Dead**
The tree in the red box appears to be dead because it lacks the lush green foliage that characterizes the healthy trees surrounding it.

Figure 22: A question case of the growth status recognition task with responses from GPT-4o and InternVL2-26B.

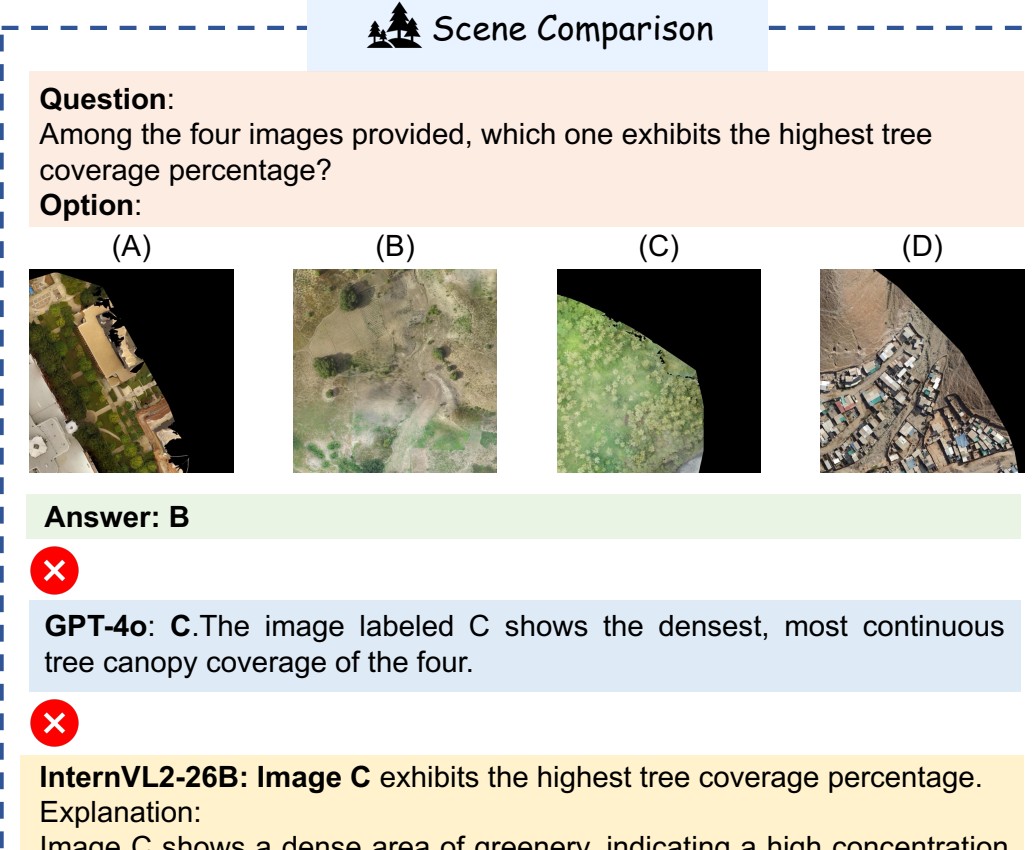

**🌲 Scene Comparison**

**Question**:
Among the four images provided, which one exhibits the highest tree coverage percentage?
**Option**:

(A)          (B)          (C)          (D)

**Answer: B**

❌

**GPT-4o**: **C**.The image labeled C shows the densest, most continuous tree canopy coverage of the four.

❌

**InternVL2-26B: Image C** exhibits the highest tree coverage percentage.
Explanation:
Image C shows a dense area of greenery, indicating a high concentration of trees and vegetation compared to the other images. The other images (A, B, and D) have less green coverage, with more visible structures, pathways, and barren land.

Figure 23: A question case of the scene comparison task with responses from GPT-4o and InternVL2-26B.

## 🌲 Scene Comparison

**Question**:
Which image shows the highest number of individually distinguishable trees?
**Option**:

| (A) | (B) | (C) | (D) |
|---|---|---|---|

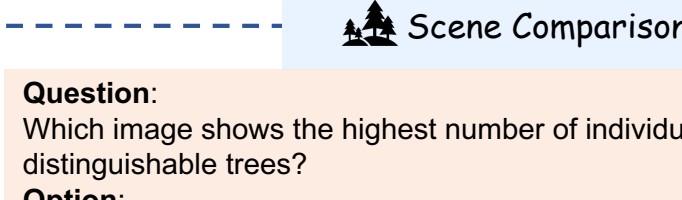

**Answer:** A

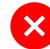

**GPT-4o**: **B**.The image labeled B shows the greatest number of individually distinguishable trees

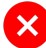

**InternVL2-26B**：**Image B** shows the highest number of individually distinguishable trees.
Explanation:
Image A shows a dense area with many trees, but they are not individually distinguishable due to the close proximity and overlapping foliage. Image B displays a more open area with trees spaced apart, making it easier to distinguish individual trees. Image C has a mix of vegetation, but the trees are not as clearly separated as in Image B. Image D shows a grassy area with very few trees, making it the least likely to have a high number of distinguishable trees.

Figure 24: A question case of the scene comparison task with responses from GPT-4o and InternVL2-26B.

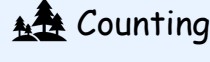 Counting

**Question**:
How many rice leaf roller are in this image?

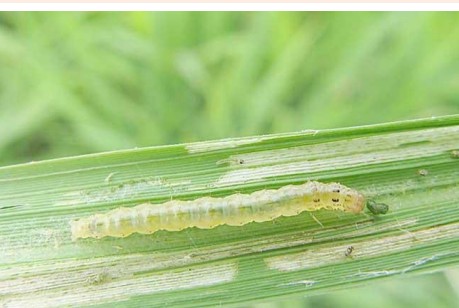

**Answer: 1**

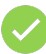

**GPT-4o**: There is a **single** rice leaf-roller caterpillar visible in the image.

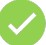

**InternVL2-26B**：There is **one** rice leaf roller in this image.
The image shows a single caterpillar, which is the larval stage of the rice leaf roller, a pest known for damaging rice crops. The caterpillar is visible on a rice leaf, which has characteristic damage marks caused by the feeding activity of the pest.

Figure 25: A question case of the counting task with responses from GPT-4o and InternVL2-26B.

### 🏕 Counting

**Question**:
How many mangos are there?

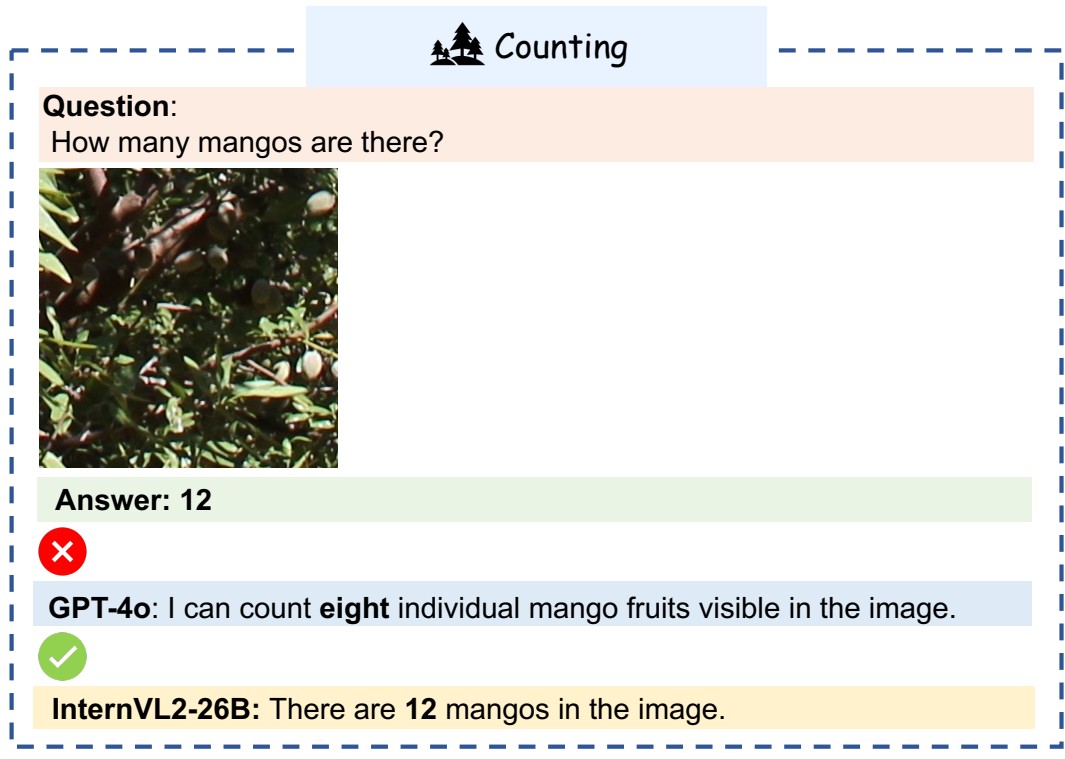

**Answer: 12**

❌

**GPT-4o**: I can count **eight** individual mango fruits visible in the image.

✅

**InternVL2-26B:** There are **12** mangos in the image.

Figure 26: A question case of the counting task with responses from GPT-4o and InternVL2-26B.

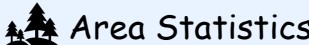 **Area Statistics**

**Question**:
What is the tree coverage percentage within the image area?
**Option**:
(A) 70-80%                    (B) 30-40%
(C) 80-90%                    (D) 50-60%

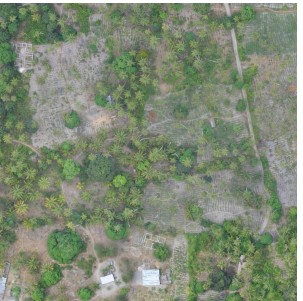

**Answer: D**

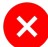

**GPT-4o**: A.The image shows an area with a significant amount of tree coverage, including areas with both dense trees and sparse patches. Based on visual analysis, I would estimate the tree coverage to be around 70-80% of the total area.Thus, the correct option is:"A": "70-80%".

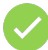

**InternVL2-26B:** The tree coverage percentage within the image area is **"D"**: "50-60%".
The image shows a mix of dense tree clusters and open, cleared areas. The trees are not uniformly distributed, and there are significant patches of land that appear to be cleared or sparsely vegetated. This suggests that approximately half of the area is covered with trees.

Figure 27: A question case of the area statistics task with responses from GPT-4o and InternVL2-26B.

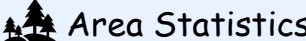 **Area Statistics**

**Question**:
What is the proportion interval of farmland area?
**Option**:
(A) 10%~20%          (B) 0%~10%
(C) 30%~40%          (D) 80%~90%

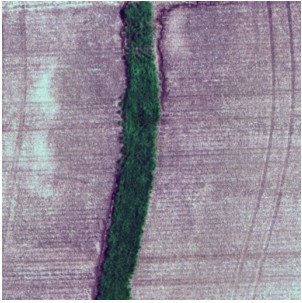

**Answer: D**

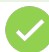

**GPT-4o**: **D**.The vast majority of the scene is tilled field (the beige area), with only a narrow green shelterbelt down the middle. Visually, the fields cover roughly 85% of the image, so the correct interval is 80%–90%

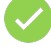

**InternVL2-26B**: The image shows a large area of farmland with a small strip of greenery. The farmland occupies most of the image.
**Answer: "D": "80%~90%"**

Figure 28: A question case of the area statistics task with responses from GPT-4o and InternVL2-26B.

## 🧠 Visual Prompt Reasoning

**Question**:
What is the crown diameter of the tree in the red box in the image when each pixel unit is 0.08m?
**Option**:
(A) 8.96m          (B) 3.52m
(C) 4.88m          (D) 13.60m

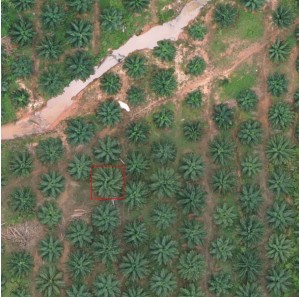

**Answer:** A

✅

**GPT-4o**: **(A) 8.96 m**. To measure the crown diameter, we detect the red-box edges in the image and find the box spans about 112 pixels across. At 0.08 m per pixel that gives.

❌

**InternVL2-26B**： Count the number of pixels across the crown of the tree in the red box.Multiply the number of pixels by 0.08m to get the diameter in meters.Assuming the crown spans approximately 60 pixels (as an example): Crowndiameter = 60 pixels × 0.08 m/pixel = 4.8m, Given the options, the closest answer is:**(C): 4.88m**

Figure 29: A question case of the visual prompt reasoning task with responses from GPT-4o and InternVL2-26B.

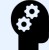

## 🧠 Visual Prompt Reasoning

**Question**:
How many tassels in the Top-Left region of the image?

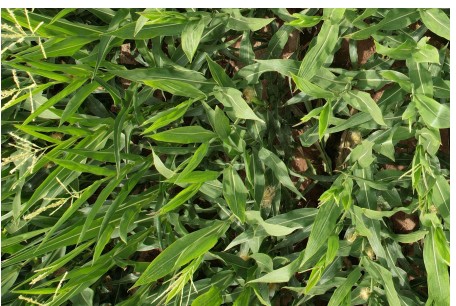

**Answer:** 2

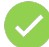

**GPT-4o**: **two**. In the top-left quadrant of the photo you can see exactly two distinct tassels.

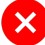

**InternVL2-26B**：There are **three** tassels in the top-left region of the image.

Figure 30: A question case of the visual prompt reasoning task with responses from GPT-4o and InternVL2-26B.

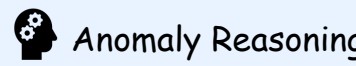

**Anomaly Reasoning**

**Question**:
Which anomalous area is in this image of farmland?
**Option**:
(A) cloud_shadow      (B) double_plant      (C) planter_skip
(D) standing_water      (E) waterway      (F) weed_cluster
(G) No anomaly

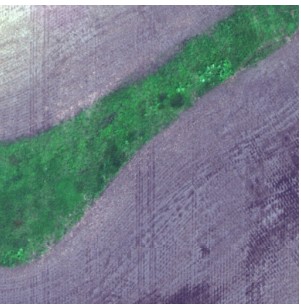

**Answer:** E

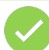

**GPT-4o**: **E**. waterway. The feature here is a vegetated drainage channel (a waterway) cutting through the field. It's not a planting error or a shadow, but rather the strip of natural vegetation along a drainage swale.

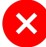

**InternVL2-26B：(F): weed_cluster**
The green area in the image appears denser and more irregular compared to the surrounding farmland, indicating a cluster of weeds.

Figure 31: A question case of the anomaly reasoning task with responses from GPT-4o and InternVL2-26B.

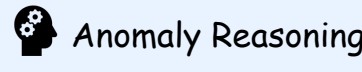

**Anomaly Reasoning**

**Question**:
Which image contains a pest that causes yield reduction in Citrus?
**Option**:

| (A) | (B) | (C) | (D) |

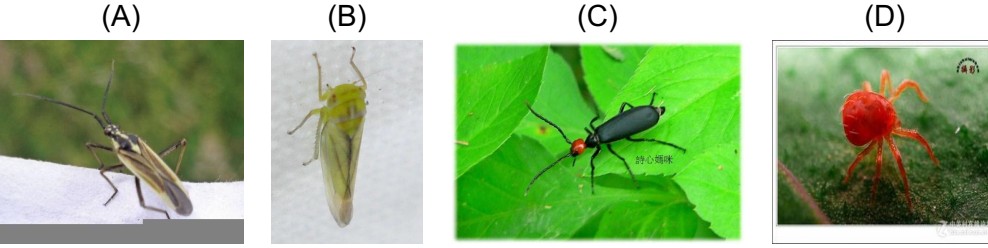

**Answer:** D

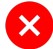

**GPT-4o**: **B**. In these four images, the pest responsible for the reduced yield of citrus crops is the image in the upper right corner (labeled B) . This image shows a citrus fruit scale (such as the citrus clownfly) that has a significant impact on the health and yield of citrus plants. The bugs in the other images (A, C, D) had relatively little effect on the citrus.

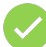

**InternVL2-26B**：**Image D** contains a pest that causes yield reduction in Citrus.
The pest in image D is a red mite, which can cause significant damage to citrus plants by feeding on the sap of the leaves, leading to reduced photosynthesis and yield.

Figure 32: A question case of the anomaly reasoning task with responses from GPT-4o and InternVL2-26B.

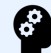 **Climate Type Reasoning**

**Question**:
According to the picture, the ecosystem belongs to which of the following options of climate zones?
**Option**:
(A) Arctic Zone  (B) Northern Temperate Zone
(C) Tropical Zone  (D) Southern Temperate Zone
(E) Antarctic Zone

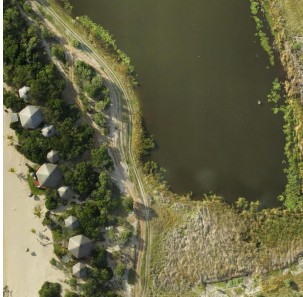

**Answer:** D

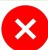

**GPT-4o**: **C**: Tropical Zone. The picture shows an ecosystem surrounded by sandy beaches, vegetated areas and water bodies, features that suggest the region may be in a tropical climate.

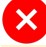

**InternVL2-26B: (C): Tropical Zone**
The image shows a lush, green landscape with a body of water and structures that resemble tropical architecture, suggesting a tropical climate zone.

Figure 33: A question case of the climate type reasoning task with responses from GPT-4o and InternVL2-26B.

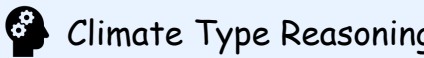 Climate Type Reasoning

**Question**:
Based on the image characteristics, what is the primary vegetation type of the ecosystem?
**Option**:
(A) Puget lowland forests
(B) Mindanao-Eastern Visayas rain forests
(C) Guinean forest-savanna mosaic
(D) Po Basin mixed forests

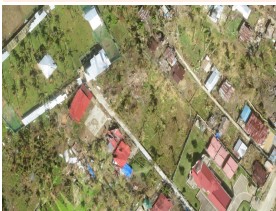

**Answer: B**

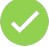

**GPT-4o**: **B**. Based on the characteristics of the ecosystem shown in the image, a large number of palm trees and other tropical plants can be seen, indicating a tropical rainforest ecosystem. Given the content of the options, the most likely options are:B: Mindanao-Eastern Visayas rain forestsThis option matches the characteristics of the tropics and is therefore likely to be correct.

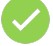

**InternVL2-26B：(B)**: Mindanao-Eastern Visayas rain forests
The image shows a tropical environment with dense vegetation and palm trees, which is characteristic of rain forests found in regions like Mindanao and the Eastern Visayas.

Figure 34: A question case of the climate type reasoning task with responses from GPT-4o and InternVL2-26B.

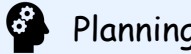

**Planning**

**Question**:
Please use 3 words or phrases to describe the quantity level, area level, and coverage level of cultivated land in the whole image.
**Option**:
quantity level:  'A small amount', 'Relatively few', 'Moderate', 'Relatively many', 'A large amount'
area level:  'Extremely limited', 'Not large', 'Relatively reasonable', 'Relatively vast', 'Extremely vast'
coverage level:  'Extremely low', 'Relatively low', 'Normal', 'Relatively high', 'Extremely high'

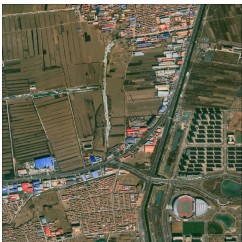

**Answer:** A large amount, Extremely vast, Normal

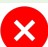

**GPT-4o**: For the image you provided, the cultivated land can be described as follows: Quantity level: **"Moderate"** Area level: **"Relatively vast"** Coverage level: **"Relatively high"**

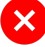

**InternVL2-26B**：
Quantity level: **Relatively many**
Area level: **Relatively vast**
Coverage level: **Relatively high**

Figure 35: A question case of the planning task with responses from GPT-4o and InternVL2-26B.

**Question**:
Based on weed coverage (0.2%) and weed-crop ratio (0.0:1), select the appropriate weeding management measure:
**Option**:
(A) Emergency weeding (coverage≥30% or ratio≥1.0)
(B) Targeted weeding (coverage 20-30% and ratio≥0.5)
(C) Monitoring required (coverage 20-30%)
(D) Preventive weeding (ratio≥0.5 and coverage<20%)
(E) No action needed

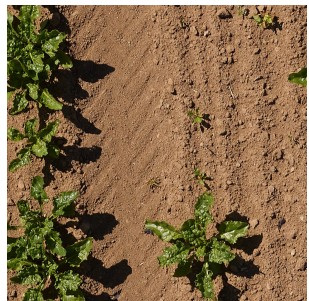

**Answer:** E

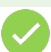

**GPT-4o**:E: No action needed.Based on the weed coverage of 0.2% and a weed-crop ratio of 0.0:1, the appropriate weeding management easure would be:**E: No action needed.**

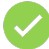

**InternVL2-26B**：**(E): No action needed**
Explanation: The weed coverage is 0.2%, which is well below the 20-30% range, and the weed-crop ratio is 0.0:1, indicating very low weed presence. Therefore, no action is needed.

Figure 36: A question case of the planning task with responses from GPT-4o and InternVL2-26B.

