# OpenReview forum: "Can Large Multimodal Models Understand Agricultural Scenes? Benchmarking with AgroMind"
_NeurIPS.cc/2025/Datasets_and_Benchmarks_Track — NeurIPS 2025 Datasets and Benchmarks Track poster_

### Official Review · Reviewer_yGqA · 2025-06-13

**Rating:** 6
**Confidence:** 5

**Summary:**

This paper presents AgroMind, a comprehensive benchmark designed to evaluate the capabilities of large multimodal models (LMMs) in agricultural remote sensing (RS) scenarios, filling the gap in the evaluation of LMMs applied to specialized domains. Unlike existing benchmarks that often suffer from limited scene diversity and overly simplified tasks, AgroMind introduces a multi-dimensional framework encompassing four core task dimensions including spatial perception, object understanding, scene understanding and scene reasoning across 13 task types. The benchmark is built from a curated dataset of 15,556 images and 25,026 question-answer pairs, derived from eight public and one private dataset, covering a wide range of real-world agricultural scenarios. A comprehensive evaluation of 21 LMMs revealing significant challenges faced by current models, particularly in fine-grained perception and spatial reasoning. In summary, this work shows a strong standard in topic selection, data construction, task challenge and analysis depth, which is a highly impactful research contribution in the field. I recommend acceptance and AgroMind is expected to be a cornerstone for future work.

**Additional Feedback:**

1) Some statistics should be made consistent between the main text and the appendix, such as scene categorization.
2) Further analysis should be conducted to better understand the possible reasons behind LMMs' failure cases in specific task types.

**Dataset Code Accessibility:**

Yes

**Dataset Code Comments:**

The provided link includes all datasets in usable formats, along with a well-documented README file. The released Python code constitutes a complete evaluation framework, including model inference and answer evaluation procedures.

**Ethical Considerations:**

No, there are no or only very minor ethics concerns

**Final Justification:**

The authors have effectively addressed my concerns regarding benchmark construction, dataset diversity, and evaluation methodologies. Their justification for hierarchical categorization, handling of open-ended questions, and plans for geographic expansion strengthen the work's robustness. The responses to other reviewers on temporal dynamics and multimodal model integration were also convincing. AgroMind represents a significant contribution that will serve as a foundational resource in agricultural remote sensing research. I recommend acceptance.

**Limitations Weaknesses:**

1) Section 4.2 lacks detail. The benchmark construction process should be explained more clearly.
2) There is inconsistency in scene categorization between Figure 3 and Figure 8: Figure 3 shows 5 scene types, while Figure 8 presents 9.
3) There is no analysis or guarantee that public training datasets used for LMMs do not overlap with AgroMind’s data.
4) The distribution of tasks in the benchmark is uneven: some tasks, such as Object Classification, are heavily overrepresented, while others, like Boundary Detection, have far fewer samples.
5) Some tasks are too templated (e.g., 3×3 grid Spatial Localization), possibly hiding model weaknesses in detailed tasks.
6) Other questions:
   1) Why use cosine similarity for open-ended questions, and how is the threshold set? Any expert comparison?
   2) For counting, some answers are too precise (e.g., 6.96m). Is such precision needed, or is there an error margin?
   3) Multi-image questions are concatenated into one image, but real tasks often need cross-image comparison. Does this weaken evaluation?
   4) Random guess accuracy is ~50% for judgement, but only 1.68% for counting. Was any weighting used to avoid “easy to guess” question types inflating the results?
   5) Are there certain regions or countries that are overrepresented or underrepresented in the dataset, and how might this potential geographic imbalance affect the evaluation results?

**Strengths Contributions:**

1) The paper targets a high-impact but underexplored domain, agricultural intelligence, where existing LMM benchmarks exhibit insufficient scene diversity in the dataset and oversimplified task design. AgroMind fills a critical gap in the LMMs community by providing the comprehensive benchmark for agricultural scene understanding and reasoning.
2) AgroMind defines 13 agriculturally meaningful tasks, which go beyond low-level perception and include reasoning, fine-grained recognition, and spatial understanding. This is a significant improvement over prior benchmarks that are limited to classification or segmentation. Tasks such as Spatial Relationship, Anomaly Reasoning, and Planning reflect real-world decision-making needs and challenge LMMs on both perception and reasoning capabilities.
3) The dataset features diverse scenes, multiple sensor types, and task dimensions, covering a broad range of real-world agricultural RS scenarios. The use of multimodal data (text and images) provides a solid foundation for evaluating the understanding of LMMs.
From data collection, normalization, annotation, and QA generation, the data pipeline is rigorous and reproducible, which ensures data quality and research reproducibility.
4) The experimental section is very detailed. The proposed benchmark is used to evaluate 21 LMMs, including both open-source and closed-source models with different parameter sizes, and uses human and random guess as baselines. The results are also analyzed with respect to the characteristics of agricultural scenarios. The benchmark serves both as a performance evaluation tool and a diagnostic framework for identifying LMM weaknesses in domain-specific cognition.

---

> ### Author Rebuttal · Authors · 2025-07-26
>
> We deeply appreciate the reviewer’s thoughtful feedback and the acknowledgment of AgroMind’s significant contribution to agricultural remote sensing evaluation. Below, we address each of your comments in detail, and clarify all concerns raised.
>
> **Q1: Clarify benchmark construction process.**
>
> A: In fact, we have comprehensively **documented the construction pipeline in Appendices B and C**, ensuring full reproducibility and traceability. Specifically, Appendix B.2 outlines the preprocessing protocols for multi-source data, including TIFF-to-PNG/JPG conversion, randomized parcel cropping for geospatial images, instance-level statistical analysis. Appendix C further details: *Human annotation and participation standards* (C.1), *Model inference and computational resources* (C.2), *Question difficulty stratification* (C.3), *Evaluation protocols per question type*, including semantic similarity thresholding via Sentence-BERT for open-ended answers (C.4). Collectively, this documentation captures the end-to-end benchmark pipeline, covering QA generation, model inference, and human-aided quality control. We will add **explicit cross-references to the appendix** in the main text to improve clarity and guide readers to detailed pipeline.
>
> **Q2: Inconsistency between Fig. 3 and Fig. 8.**
>
> A: Sorry for the misunderstanding. The discrepancy between Fig. 3 and Fig. 8 results from a **hierarchical taxonomy** rather than inconsistency. Fig. 3 aggregates scenes into 5 macro-categories to **visualize dataset structure**, while Fig. 8 refines them into 9 subclasses to microscopically **validate task diversity**. For instance, "Crop Monitoring" in Figure 3 is divided into Crop, Maize Tassel, and Fruit in Figure 8, whereas "Tree Analysis" splits into Tree (general forestry) and OilPalm. This layered approach maintains framework conciseness while using subclass visualization (Fig. 8 bar chart) to highlight agricultural scenario richness. We will clarify this hierarchical relationship in the revised caption of Fig. 3 and Fig. 8 to avoid confusion.
>
> **Q3: Overlap with public training datasets.**
>
> A: Potential overlaps between public training datasets and AgroMind’s evaluation data **do not compromise the validity of our benchmark**. First, AgroMind is fundamentally differ from coarse-grained annotations in source datasets such as class labels. During training, models only access raw annotations, whereas AgroMind requires solving complex domain-specific problems (e.g., "calculating crown diameter from coordinates" or "inferring crop types using climatic features"), demanding deeper agricultural knowledge and reasoning. Even if identical images appear in training, models cannot rely on simple pattern matching and must leverage contextual understanding to address AgroMind’s unique queries. This design guarantees discriminative evaluation, with any overlap having negligible impact on robustness.
>
> **Q4: Imbalanced task distribution.**
>
> A: To address task distribution imbalance, we conducted **sample expansion** of QA pairs. We added task types with fewer samples to Fig. 4, mainly the task volume of the 3 subtasks: **Boundary Detection, Spatial Relationship, and Spatial Localization**. The proportions of the 4 major tasks then became close to **5:7:7:7**. This enhancement ensures that models are evaluated across a **comprehensive range of task types**, supporting more nuanced performance assessment.
>
> **Q5: Templated question types.**
>
> A: We appreciate this point and would like to clarify the rationale behind task design. Actually, there is no such concern. The **3×3 grid spatial localization** task is intended as a **coarse-grained, foundational spatial reasoning** benchmark, similar to a heatmap prediction or weakly supervised location estimation. To address concerns regarding granularity, AgroMind also includes **high-resolution plot boundary detection** tasks, which require the model to localize **precise bounding coordinates** or extract fine-grained vector boundaries—a detailed and challenging task. By combining both types of localization, the benchmark achieves a **layered evaluation protocol**:
> - Grid-based localization assesses macro spatial understanding,
> - Plot boundary tasks evaluate detailed spatial precision.
> They form a multidimensional localization capability assessment system that enables the benchmark to measure the model's spatial understanding capabilities more comprehensively and accurately.
>
> **Q6-1: Cosine similarity for open-ended questions.**
>
> A: We sincerely appreciate the reviewer's feedback on the evaluation methodology for open-ended answers. Below, we provide a detailed clarification of our evaluation process, including the **evaluation criteria**, the involved model, and the **threshold-setting process**.
>
> All open-ended questions have clear and fixed reference answers, which are designed or verified by domain experts and serve as the ground truth for **semantic comparison**. Specifically, we use the **Paraphrase-MiniLM-L6-V2** model [1], a lightweight and high-performing sentence embedding model from the Sentence-BERT family, to compute **cosine similarity** between each model-generated answer and the reference answer. Paraphrase-MiniLM-L6-V2 has been extensively trained on large-scale paraphrase datasets and is capable of recognizing semantic equivalence even across diverse linguistic expressions — making it particularly suitable for evaluating semantic consistency in open-ended agricultural QA.
>
> We define an open-ended prediction as correct only when its **semantic similarity score** with the reference answer **exceeds a fixed threshold**. Through **preliminary experiments on a dataset subset**, we determined the **threshold as 0.60**, under which all models achieve reasonable accuracy.
>
> **References:**
>
> [1] Reimers N, Gurevych I. Sentence-bert: Sentence embeddings using siamese bert-networks[J]. arXiv preprint arXiv:1908.10084, 2019.
>
> **Q6-2: Answer’s precision.**
>
> A: The benchmark **does not require decimal-level precision** for counting tasks. Both the predicted and ground-truth answers are rounded to the nearest integer before comparison. The evaluation focuses on numerically reasonable estimations rather than minor rounding variations, aligning better with practical agricultural applications.
>
> **Q6-3: Evaluation of multi-image questions.**
>
> A: Concatenating images may lower performance for some models. However, since **some models only support single-image input (eg. TinyLLaVA, LLaVA-NeXT-7B-Mistral, XComposler2-4KHD, InstructBLIP-Vicuna-7B, Idefics-2-8b)**, we concatenate all images to ensure a fair comparison across all models.
>
> **Q6-4: Random guess accuracy.**
>
> A: Thank you for highlighting this issue. We acknowledge the variance in baseline guessing difficulty across task types. However, we intentionally **do not apply weighting** during overall accuracy computation. Our primary goal is to benchmark relative **model performance across diverse tasks** under identical aggregation metrics, not to estimate real-world task importance or difficulty distributions. What’s more, introducing post-hoc weighting risks introducing subjective bias and undermining evaluation consistency. We report **per-task accuracy breakdowns** in the results section (and supplementary materials), allowing users to interpret model performance with awareness of underlying task difficulty.
>
> **Q6-5: Potential geographic imbalance.**
>
> A: Thank you for raising this important concern. AgroMind, intentionally **covers a diverse range of regions**, including China, the United States, the European Union, Brazil, and parts of Africa, to reflect global agricultural diversity in terms of crops, climate, and land use patterns. We acknowledge that some regions (e.g., certain African and Southeast Asian countries) are relatively underrepresented due to limitations in publicly available, high-resolution multimodal agricultural data. However, this does not affect the evaluation of the model performance. To enhance transparency, we will include **geographical coverage maps** of datasets in Appendix B.1, where the number of samples per region is visually indicated. Furthermore, we explicitly discuss the implications of this imbalance and its potential impact on fairness in Appendix C.5, and we plan to address this limitation by continuously expanding the dataset with better-balanced regional representation in future releases.
>
> **Q7: Minor issues: consistency of statistics, better analysis of LMM failure modes.**
>
> A: Thanks for your careful review and helpful suggestions. We have corrected all inconsistencies between the main text and the appendix (e.g., scene categories, task numbers). We will add a detailed error taxonomy and failure analysis in Appendix C.5, highlighting representative LMM failure cases across perception and reasoning tasks.
>
> Once again, we sincerely thank *Reviewer yGqA* for the thoughtful and encouraging feedback. We believe your suggestions have significantly improved the clarity and rigor of our work. We are excited about the potential of AgroMind to become a valuable asset for both the LMM and agricultural remote sensing communities. We will **address all the above issues in the final manuscript** if the paper is accepted. We believe that our revisions and responses demonstrate the strengthened value of this work.

---

> > ### Comment · Reviewer_yGqA · 2025-08-02
> >
> > The authors have effectively addressed my concerns regarding benchmark construction, dataset diversity, and evaluation methodologies. Their justification for hierarchical categorization, handling of open-ended questions, and plans for geographic expansion strengthen the work's robustness. The responses to other reviewers on temporal dynamics and multimodal model integration were also convincing. AgroMind represents a significant contribution that will serve as a foundational resource in agricultural remote sensing research. I recommend acceptance.

---

> > > ### Author Response · Authors · 2025-08-03
> > >
> > > Thanks for your feedback. We sincerely appreciate your recognition of our work and the constructive suggestions that have improved our paper. We also welcome your participation in further discussions regarding the General Comments.

---

### Official Review · Reviewer_GvXT · 2025-06-26

**Rating:** 4
**Confidence:** 3

**Summary:**

This paper proposes a benchmark for agricultural remote sensing, AgroMind. Diverse from existing benchmark focusing on nature images, AgroMind incorporates images from remote sensing covering four task dimensions: spatial perception, object understanding, scene understanding, and scene reasoning. 18 open-source LMMs and 3 closed-source models are evaluated on AgroMind, with some significant findings.

**Dataset Code Accessibility:**

Partly

**Dataset Code Comments:**

The hugging face link does not include a proper readme to illustrate the usage and describe each file. I cannot find the QA pairs to evaluate the quality.

**Ethical Considerations:**

No, there are no or only very minor ethics concerns

**Final Justification:**

The authors' rebuttal addresses some of my concerns. Overall, I don't oppose to the acceptance, so I will raise the score above the borderline.

**Limitations Weaknesses:**

1. While the authors use remote sensing as the motivation, the final dataset actually contains a large proportion of natural images.
2. The intersection between natural and remote sensing is minimal. Natural images for agriculture are very diffierent from remote sensing ones. If the knowledge in nature images does not improve the understanding of remote sensing, the integration of the two scenes could be less meaningful. It seems like the authors just integrate several irrelevant things without thinking about the relation between them.
3. The datasets contain many augmented images to make it seem large.
4. Some of the QA pairs are generated by LMM. While the authors mention they have a quality control, it is unclear how, and in what proportion, the quality control is carried out.

**Strengths Contributions:**

1. The topic of creating benchmark for LLMs in agriculture is worth exploring and highly related to the conference.
2. The proposed benchmark integrates 8 publich datasets and 1 private dataset to create a dataset larger then existing ones, which has potential impact.

---

> ### Author Rebuttal · Authors · 2025-07-29
>
> Thanks for your efforts and comments. We appreciate your recognition of the benchmark’s significance and potential impact. Below, we provide detailed point-by-point responses and clarify all concerns raised.
>
> **Q1: Proportion of natural images**
>
> A: We understand your concern and would like to clarify our definition and rationale as follows.
>
> **1. Broader definition of remote sensing:**
>
> The term *remote sensing* (RS) in our work is adopted according to its **broad definition** widely accepted in geoscience and precision agriculture communities. RS is not limited to satellite or aerial imagery but refers to **any non-contact observation method** used to capture data from a distance [1-2]. This includes UAVs, multispectral/hyperspectral sensors, proximal platforms, and even **handheld or smartphone-based imaging systems** commonly deployed in smart farming practices. In this context, **camera-captured imagery** collected by human observers belongs to the **proximal RS**, particularly under plant phenotyping and field-scale monitoring scenarios.
>
> **2. Plan to enhance the narrow remote sensing coverage:**
>
> To strengthen the agricultural RS focus, we plan to **expand the dataset by integrating the CropHarvest** [3] open-source agricultural RS dataset (90,480 data points, paired with Sentinel-2, Sentinel-1 and SRTM). The updated dataset will increase the proportion of *Narrow RS* images to over 70% and introduce multi-faceted tasks (e.g., crop classification, yield prediction) to enhance its scientific utility. The added data will cover major global agricultural regions (e.g., Kenya, Brazil), leveraging high-resolution **multispectral optical images, SAR and topographic data** to monitor soil moisture and crop growth dynamics.
>
> **References:**
>
> [1] Avery T E, Berlin G L. Fundamentals of Remote Sensing and Airphoto Interpretation [M]. 5th ed. New York: Macmillan; Maxwell Macmillan Canada, 1992.
>
> [2] Bhatta B. Research methods in remote sensing[M]. Berlin: Springer, 2013.
>
> [3] Tseng G, Zvonkov I, Nakalembe C L, et al. Cropharvest: A global dataset for crop-type classification[C]//Thirty-fifth Conference on Neural Information Processing Systems Datasets and Benchmarks Track (Round 2). 2021.
>
> **Q2: Intersection between natural and remote sensing images**
>
> A: First, we fully agree that natural camera and satellite images differ in format, resolution, and perspective. However, the difference is **not a drawback but an advantage**, as they capture **complementary spatial and semantic scales**:
> - Satellite or UAV provides macro-level field context such as crop type, land cover, and drought risk.
> - Natural images offer fine-grained object-level features, such as pest status, leaf curling, and chlorosis.
>
> The **multi-scale integration mirrors real-world scenario**, where aerial data highlights a potential issue and ground images confirm it.
>
> Furthermore, our introduction of natural images is not motivated by image-level fusion, but based on **a unified design at the semantic level** of agricultural RS tasks. For example, *"Is the crop drought affected?"* This problem can be judged by analyzing NDVI index from satellite images, or observing leaf curl and color change from close-range images. Although they come from different sources, they together constitute **a multi-view reasoning task** for realistic problems, which is an urgent ability dimension to be covered in the current LMMs evaluation.
>
> **Q3: Augmented images make datasets seem large**
>
> A: In fact, we **do not intentionally use image augmentation** (e.g., flipping, rotation) to artificially enlarge the dataset. Each image corresponds to a real observation from public agricultural datasets. To ensure transparency, we have conducted a thorough manual inspection and will **remove any accidentally repeated or overly similar samples**.
>
> What’s more, the dataset may **appear large in terms of storage or image dimensions** because RS images typically have higher spatial resolution (e.g., 2048×2048 or larger), especially for UAV or satellite imagery. A single image tile may cover a large field and be significantly larger than typical web-crawled natural images. Thus, the per-image file size is larger, but the **actual number of unique images is not artificially inflated.**
>
> We will **release an updated dataset version** with deduplicated samples and clearly documented statistics. We will also revise the manuscript to clarify these dataset characteristics and improvements.
>
> **Q4: Quality control**
>
> A: We would like to present a supplementary explanation on the quality control process. Contrary to the impression that most QA pairs are directly generated by LMMs, we perfer to emphasize that the majority of questions were initially derived based on **domain experts' design**, or **extracted directly from existing image-level labels** and metadata provided by the original datasets. LMMs were used only to rephrase or **assist in converting structured labels to fluent questions**, under strict human supervision.
>
> To improve the quality of the dataset, we implement a strict quality control process to **manually validate each QA pair**, which included the following components.
> 1. Guideline preparation: Before the annotation phase, a team of domain experts (in remote sensing, and agricultural data science) designed detailed quality control guidelines, covering principles such as: (a) **remove ambiguous, irrelevant, or visually mismatched questions**; (b) ensure each referring expression clearly and explicitly maps to a visual target or concept; and (c) avoid including questions that inherently contain their own answers.
> 2. Verification: Each QA pair in the dataset was **individually verified** by at least one trained human annotator with a strong background in the target domain. The average verification time was approximately 150 seconds per image, resulting in **over 1,042 total hours**.
> 3. Cross-validation: To further improve reliability, we conducted a **double-blind verification** on a randomly selected subset of 2,000 QA pairs. The systematic process can identify recurring annotation inconsistencies, resolve labeling discrepancies, and further strengthen dataset reliability.
> 4. Examples of manual correction: Below we provide two cases to illustrate how human intervention improved QA quality.
> - Case 1 (Counting task validation)
> ```
> {
>   "image_path": "./corn/images/image_0210.jpg",
>   "question": "How many tassels are there in this image?",
>   "initial_answer": "4",
>   "answer": "5"// Corrected to "5" after identifying a partially hidden tassel
> }
> ```
> - Case 2 (Disease diagnosis validation)
> ```
> {
>   "image_path": "./Plant_Disease/Apple___Cedar_apple_rust/809c7104-6f46-4117-8c8c-6d5491b2178d___FREC_C.Rust3839_90deg.JPG",
>   "question": "Is the crop in the image healthy? If not, what disease is affecting it, and what effective measures should be taken to address the issue?",
>   "initial_answer": "The apple plant looks unhealthy and has spots. It might be a kind of rust disease. You can try pruning or use some fungicide to treat it.",
>   "answer": "The apple is affected by Cedar apple rust. Recommended treatment: 1. Spray with 10% difenoconazole WG 1500× after blooming; 2. Foliar spray with 0.3% urea to improve resistance; 3. Prune dense branches."
>   // Revised for accuracy and standardized according to agricultural guidelines
> }
> ```
> **Q5: Dataset code, README file and QA pairs**
>
> A: While we had already provided such resources on our GitHub repository, we agree the importance of having everything integrated and accessible directly via Hugging Face. To address this, we will **update the Hugging Face** page to include a clear **README.md** file that outlines the dataset structure, describes each file, provides loading instructions, and presents representative QA examples across task types. We will **provide representative QA pair examples** from different task types (e.g., object understanding, scene reasoning) in the README and as a standalone JSON file.
>
> Additionally, we provide many visual examples in the **Appendix E**, covering representative QA cases for each of the 13 task types. The case studies demonstrate the dataset's diversity in agricultural scenarios and annotation quality. We hope the additions sufficiently address the reviewer's concern and demonstrate the value and completeness of the benchmark.
>
> - The main framework of README.md
> ```
> • Project Description: A brief introduction outlining the goals and scope of the AgroMind.
> • Access Links: Links to the paper, GitHub repository, and Hugging Face dataset.
> • Data Structure: Files are grouped into thematic folders such as Crop and Pest, along with a separate 'QA' directory that contains QA pairs for each task.
> • QA Format Specification: Each QA file is in JSON format, containing structured items with fields such as image_path, question, answer, and task-level identifiers. The item is structured as follows:
> {
>   "image_path": "path/to/image",
>   "type_id": question_format_type,
>   "item_id": "id",
>   "level1_id": "main_category", // Top-level task dimension
>   "level2_id": "sub_category", // Task subtype
>   "level3_id": "specific_task", // Detailed task type
>   "question": "query_text", // Natural language question
>   "options": ["A", ...], // Answer choices (when applicable)
>   "answer": "correct_response" // Ground truth answer
> }
> • Usage Instructions: Guide users on how to use them for LMMs evaluation.
> • Citation & License: Details for proper academic citation and licensing terms.
> ```
> Once again, we sincerely thank *Reviewer GvXT* for the thoughtful and encouraging feedback. We have carefully **addressed each of your concerns** through clarifications, dataset improvements, and documentation updates. We hope you recognize the practical value and solid foundation of this work, and support its broader influence in the field of AI for agriculture. We will **address all the above issues in the final manuscript** if the paper is accepted.

---

> > ### Comment · Reviewer_GvXT · 2025-08-02
> >
> > Thanks for your response, which addresses some of my concerns.
> >
> > In fact, I downloaded some of the data, and found that 1) Many images are just natural images the same as existing datasets. While the authors claim that natural images are also defined as "proximal RS", it seems unconvincing to me. 2) Some images are flipped or rotated version of each other. While the authors mention they will release an updated version to include more RS images and delete repeated samples, its properties (e.g. image count) will defer from the original ones in the paper, and I won't be able to evaluate the updated version at the moment.
> >
> > I will raise the score based on other reviewers' comments if they agree that 1) Natural images are also within the definition of remote sensing as "proximal RS" to form a benchmark that claims RS as a main contribution. 2) The concern of repeated images can be addressed by the future promise that "we will release an updated dataset version".

---

> > > ### Author Response · Authors · 2025-08-03
> > > **Response to concerns on dataset scope and repeated samples**
> > >
> > > Thank you very much for your detailed follow-up. We sincerely appreciate your careful inspection of the dataset. We would like to provide further clarification on remote sensing scope and dataset validity.
> > >
> > > On the definition of remote sensing, the scope of remote sensing has evolved significantly in recent years with the advancement of acquisition platforms. In addition to satellite and aerial imagery, proximal remote sensing, including images captured by UAVs, **handheld devices, and ground-based sensors**, has become an increasingly important subfield, especially in agriculture, environmental monitoring, and urban studies. In this context, **natural images collected under field conditions are consistent with the definition of proximal RS**, and including them in our benchmark reflects the diversity and realism of contemporary RS applications. The perspective is aligned with ongoing developments in the RS community, and has been supported by other researchers [1] [2]. As stated in [1], *"In precision agriculture (PA), non-destructive and non-invasive remote and proximal sensing methods have been widely used to observe crops in visible and invisible spectra."* The authors further cited *"some examples of public datasets for agriculture remote sensing at the leaf or canopy scale, e.g., CWFI [3], PDD217 [4]."*
> > >
> > > We confirm that **no intentional flipping or rotation was performed** in our dataset construction. However, after careful review, we found that a few original datasets contained potential duplicates, possibly due to **internal data augmentation by their publishers**. The 2018 AI Challenge Dataset we adopted for the pest and disease task included a number of augmented images, mainly flipped or rotated copies that were already present in the original data source. The augmentations were not introduced during our benchmark construction, but we recognize their potential impact on the fairness and integrity of the dataset. As a result, we have **removed all identified augmented images and their corresponding QA pairs. In total, 627 camera images with flipping or rotation patterns** were deleted, along with 1,189 associated QA samples. The current dataset version which can be downloaded has **completely eliminated duplicate images**. The related modifications in the paper will be updated as soon as possible.
> > >
> > > We hope that our thorough revisions and clarifications address your concerns. We kindly ask you to consider reevaluating our work, and thank you for providing us with valuable suggestions for improvement.
> > >
> > > **References:**
> > >
> > > [1] Wang D, Cao W, Zhang F, et al. A review of deep learning in multiscale agricultural sensing[J]. Remote Sensing, 2022, 14(3): 559.
> > >
> > > [2] Surendran, U., Nagakumar, K., Samuel, M.P. (2024). Remote Sensing in Precision Agriculture. In: Priyadarshan, P.M., Jain, S.M., Penna, S., Al-Khayri, J.M. (eds) Digital Agriculture. Springer, Cham.
> > >
> > > [3] Haug S, Ostermann J. A crop/weed field image dataset for the evaluation of computer vision based precision agriculture tasks[C]//European conference on computer vision. Cham: Springer International Publishing, 2014: 105-116.
> > >
> > > [4] Liu X, Min W, Mei S, et al. Plant disease recognition: A large-scale benchmark dataset and a visual region and loss reweighting approach[J]. IEEE Transactions on Image Processing, 2021, 30: 2003-2015.

---

> > > ### Author Response · Authors · 2025-08-05
> > > **Response to Reviewer GvXT**
> > >
> > > We sincerely appreciate your previous comments and the time you took to review our work in detail. We have provided additional clarification and dataset refinement to address the issues raised, and we also note that other reviewers have also acknowledged and supported our explanations. If you have any remaining concerns or suggestions, please let us know. Your feedback is important to us, and we would be glad to further discuss or clarify any remaining points.

---

> > ### Comment · Reviewer_yGqA · 2025-08-04
> >
> > I think natural images captured by a camera reasonably fall under the definition of remote sensing, which is widely accepted in precision agriculture and remote sensing communities.
> >
> > I also downloaded and reviewed the dataset. In the updated dataset, I observed no instances of duplicate or repeated images. After removing the duplicate samples (inheriting from the original datasets), all reported performances need to be updated in the final version.

---

### Official Review · Reviewer_bypw · 2025-07-01

**Rating:** 5
**Confidence:** 4

**Summary:**

The authors introduce AgroMind, a multi-scenario, multi-task benchmark for evaluating LMMs in agricultural context. It covers 13 task types across four core dimensions: spatial perception, object understanding, scene understanding, and scene reasoning. The benchmark is also used to evaluate 18 open-source and 3 closed-source LMMs.

**Additional Feedback:**

- It would be helpful to clarify the rationale for selecting Gemini Flash over the more commonly used and capable Gemini Pro.
- Given that domain-specific MLLMs are often intended for professional users, future versions of the benchmark could consider incorporating expert annotators as human baselines, as this may lead to more meaningful comparisons with model performance.

**Dataset Code Accessibility:**

Partly

**Ethical Considerations:**

No, there are no or only very minor ethics concerns

**Final Justification:**

The authors have effectively addressed my concerns regarding the evaluation of domain-specific LLMs and sensor-wise analysis, and have provided new insights in their response. I believe these additions further strengthen the contribution of AgroMind, and I am therefore willing to raise my score.

**Limitations Weaknesses:**

- One limitation of the study is that it only compares general-purpose MLLMs. Given that the benchmark is designed for agricultural remote sensing, it would be more comprehensive and representative to include domain-specific MLLMs tailored for remote sensing.
- It would be helpful to report model performance across different sensor types to examine how general-purpose MLLMs vary in their understanding of agricultural concepts from different viewpoints.

**Strengths Contributions:**

- The benchmark includes a rich set of task types that comprehensively evaluate LMMs across four core dimensions: spatial perception, object understanding, scene reasoning, and scene understanding. The inclusion of diverse question formats (e.g., image choice, text choice, and open-ended questions) further supports a well-rounded assessment of model capabilities in agricultural scenarios.
- The benchmark includes images from diverse sensor modalities (e.g., satellite, UAV, and ground-based cameras), aligning with the growing use of multi-source data in smart agriculture and enhancing its real-world applicability.

---

> ### Author Rebuttal · Authors · 2025-07-29
>
> We sincerely thank the reviewer for their detailed reading and insightful comments. We are pleased to know that you recognized the importance and technical rigor of our work. Below, we provide detailed point-by-point responses and clarify all concerns raised.
>
> **Q1: Include domain-specific LMMs tailored for remote sensing**
>
> A: Thank you for pointing this out. We fully agree with the importance of evaluating **domain-specific LMMs** in addition to general-purpose models. To address this concern, we have **included two recently proposed remote sensing-specific LMMs**, namely **GeoChat** (2024) [1] and **GeoLLaVA-8K** (2025) [2], as depicted in **Table 1**. These models were trained on well-known datasets such as NWPU-RESISC45 [3] and DOTA [4], which are primarily focused on scene classification and object detection in aerial images. However, they are not explicitly optimized for agricultural scenarios involving complex semantics like disease identification, crop reasoning, or spatiotemporal dynamics. During our evaluation, we observed that:
> - GeoChat achieves reasonable performance on basic perception tasks (e.g., object recognition) but struggles with tasks requiring scene-level reasoning or multimodal contextualization.
> - GeoLLaVA-8K performs well on visual question answering for RS images but converts all non-open-ended questions into multiple-choice formats (e.g., counting), which limits its ability to tackle free-form spatial reasoning and semantic interpretation tasks in AgroMind.
>
> We further note that while these models perform well in general remote sensing tasks, both models exhibit signs of **catastrophic forgetting** in agricultural scenarios, which may be due to their fine-tuning processes that override previously learned general-purpose capabilities, leading to a drop in performance on open-ended or high-level inference tasks.
>
> **Table 1. Performance comparison across multiple tasks and models. (SP: Spatial Perception, OU: Object Understanding, SU: Scene Understanding, SR: Scene Reasoning)**
>
> |Model|SP| | |OU| | |SU | | |SR| | | |Overall|
> |---|---|---|---|---|---|---|---|---|---|---|---|---|---|---|
> | |SL|SR|BD|OC|PDD|GSR|SC|CO|AS|VPR|AR|CTR|PL| |
> |GPT-4o|25.71|13.50|20.50|**81.62**|**49.70**|45.42|**36.81**|**33.34**|**45.75**|**34.00**|30.05|58.76|27.39|**42.98**|
> |InternVL2-26B|21.46|13.50|18.25|69.50|43.90|28.33|32.47|28.37|45.13|17.57|12.72|57.62|25.77|35.84|
> |**GeoChat**|32.79|8.33|30.00|44.26|27.00|38.33|20.73|24.05|16.56|8.57|18.32|41.86|27.50|26.15|
> |**GeoLLaVA-8K**|9.29|11.67|12.50|27.45|21.00|10.83|18.29|8.21|19.63|7.76|9.92|29.07|7.92|15.24|
>
> We have also actively explored **agricultural-specific multimodal large models**, such as **AgroGPT [5]** and **Agri-LLaVA [6]**. Unfortunately, **no LMM for agricultural** scene has been **officially open-sourced or released with inference APIs**, as of now. We thoroughly searched their corresponding papers, GitHub repositories, and public model hubs (e.g., Hugging Face), but no reproducible implementations were found. The lack of access prevented us from including them in the current evaluation. However, we will actively monitor their release status and plan to integrate them into AgroMind once they become available. The evaluation results highlight a broader issue in the domain of agriculture-focused LMMs: while there is increasing academic interest, **the absence of openly accessible, reproducible models** significantly hinders community-wide benchmarking and downstream application development. We hope our work on AgroMind can help promote future open-source releases in this space by providing a standardized evaluation protocol.
>
> **References:**
>
> [1] Kuckreja K, Danish M S, Naseer M, et al. Geochat: Grounded large vision-language model for remote sensing[C]//Proceedings of the IEEE/CVF Conference on Computer Vision and Pattern Recognition. 2024: 27831-27840.
>
> [2] Wang F, Chen M, Li Y, et al. GeoLLaVA-8K: Scaling Remote-Sensing Multimodal Large Language Models to 8K Resolution[J]. arXiv preprint arXiv:2505.21375, 2025.
>
> [3] Cheng G, Han J, Lu X. Remote sensing image scene classification: Benchmark and state of the art[J]. Proceedings of the IEEE, 2017, 105(10): 1865-1883.
>
> [4] Xia G S, Bai X, Ding J, et al. DOTA: A large-scale dataset for object detection in aerial images[C]//Proceedings of the IEEE conference on computer vision and pattern recognition. 2018: 3974-3983.
>
> [5] Awais M, Alharthi A H S A, Kumar A, et al. Agrogpt: Efficient agricultural vision-language model with expert tuning[C]//2025 IEEE/CVF Winter Conference on Applications of Computer Vision (WACV). IEEE, 2025: 5687-5696.
>
> [6] Wang L, Jin T, Yang J, et al. Agri-llava: Knowledge-infused large multimodal assistant on agricultural pests and diseases[J]. arXiv preprint arXiv:2412.02158, 2024.
>
> **Q2: Model performance across different sensor types**
>
> A: Thanks for your suggestions. We have **disaggregated model performance based on sensor types**, including images captured by **satellites, unmanned aerial vehicles (UAVs), and ground-based cameras**, as shown in **Table 2**. On UAV-captured images, where the objects are typically smaller, and the context tends to be more complex due to higher variability in altitude, angles, and resolutions, most models struggle significantly. In contrast, human participants demonstrate a notably higher understanding of these images, achieving accuracy rates up to 42.17%. However, for satellite and camera images, which generally have clearer object representations and more consistent viewpoints, the performance of human evaluators is markedly lower than that of LMMs. The discrepancy indicates that LMMs excel at interpreting more structured and standardized image inputs, whereas their performance sharply declines with UAV imagery characterized by greater complexity and variability. Therefore, further enhancements in understanding UAV-captured images remain a critical area for future research and development.
>
> **Table 2. Performance comparison with aggregated question sensors**
>
> |Model|UAV|Satellites|Camera|Overall|
> |---|---|:---:|:---:|:---:|
> |Human|42.17|28.89|34.72|34.72|
> |Random|20.00|18.88|19.38|19.25|
> |GPT-4o|28.30|34.41|50.87|42.98|
> |Gemini-1.5-Flash|31.60|31.68|49.00|41.21|
> |**Gemini-1.5-Pro**|19.50|33.01|45.85|38.99|
> |Claude-3.5-Sonnet|28.65|29.29|36.45|33.33|
> |DeepSeek-VL2-tiny|36.80|17.00|33.22|27.51|
> |DeepSeek-VL2-small|18.70|20.41|34.16|27.84|
> |TinyLLaVA|17.45|27.51|33.27|29.88|
> |InternVL2-2B|29.35|25.42|30.42|28.49|
> |InternVL2-4B|14.65|28.63|35.79|31.46|
> |XComposer2-4KHD|25.65|28.32|35.96|32.31|
> |LLaVA-NeXT-7B-Mistral|34.35|31.78|30.12|31.07|
> |LLaVA-NeXT-7B-Vicuna|18.00|18.34|29.77|24.61|
> |InstructBLIP-Vicuna-7B|18.65|14.21|19.90|17.70|
> |LLaVA-NeXT-Interleave-7B|10.35|15.30|20.62|17.83|
> |Mantis-LLaMA3-SigLIP|21.20|23.57|38.50|31.60|
> |Mantis-Idefics2|28.75|22.65|38.47|31.84|
> |LLaVA-NeXT-8B|16.95|31.55|34.61|32.07|
> |InternVL2-8B|25.35|32.17|38.08|34.88|
> |Idefics-2-8b|23.50|25.45|35.54|30.85|
> |LLaVA-NeXT-13B|17.25|29.01|35.11|31.43|
> |InternVL2-26B|17.35|31.64|41.35|35.84|
> |LLaVA-NeXT-34B|16.10|34.41|38.02|34.93|
> |**GeoChat**|17.00|22.04|30.25|26.15|
> |**GeoLLaVA-8K**|22.50|15.53|13.99|15.24|
>
> **Q3: Clarify the rationale for choosing Gemini Flash over Gemini Pro**
>
> A: Initially, we selected Gemini-1.5-Flash because it consistently outperformed Gemini-1.5-Pro on our benchmark during pilot testing. We have **included experimental results using Gemini-1.5-Pro**, as shown in **Table 3**. Interestingly, **Gemini-1.5-Pro achieved slightly lower performance** than Gemini-1.5-Flash, especially in spatial reasoning and object understanding tasks. We inference that Gemini-Pro’s broader general-purpose focus and heavier architecture may dilute its attention in domain-specific contexts such as agricultural remote sensing. Moreover, Gemini-Flash offers **faster inference speed and lower cost**, making it more practical for real-world applications in agricultural domains where scalability and accessibility are essential.
>
> **Table 3. Performance comparison across multiple tasks and models. (SP: Spatial Perception, OU: Object Understanding, SU: Scene Understanding, SR: Scene Reasoning)**
>
> |Model|SP| | |OU| | |SU | | |SR| | | |Overall|
> |---|---|---|---|---|---|---|---|---|---|---|---|---|---|---|
> | |SL|SR|BD|OC|PDD|GSR|SC|CO|AS|VPR|AR|CTR|PL| |
> |Gemini-1.5-Flash|25.00|17.50|28.75|75.42|48.05|**65.17**|34.64|29.07|39.11|33.06|23.92|**60.82**|28.06|41.21|
> |**Gemini-1.5-Pro**|31.69|28.33|30.00|73.95|53.50|30.00|35.98|27.27|40.18|17.55|19.08|68.60|28.33|38.99|
>
> **Q4: Future suggestions about expert human baselines**
>
> A: Thank you for this valuable suggestion. We are currently collaborating with **several domain experts in agriculture** and remote sensing to curate a **professional-level** question-answer dataset, which will serve as an expert human baseline in future benchmark updates. The new benchmark split will include expert-generated open-ended answers, rationales, and confidence levels, offering more **rigorous comparisons between LMMs and domain experts**. We plan to include this in the next version of AgroMind and cite this plan in *future work*. We believe that integrating professional-level baselines will provide further insights into when and why LMMs succeed or fail in real-world agricultural decision-making scenarios.
>
> Once again, we sincerely thank *Reviewer bypw* for the thoughtful and encouraging feedback. We have responded by adding domain-specific models, sensor-wise analysis, more model results, and expert baseline planning. We believe these enhancements substantially strengthen the contribution and clarity of the benchmark. We will **address all the above issues in the final manuscript** if the paper is accepted.

---

> > ### Comment · Reviewer_bypw · 2025-08-06
> >
> > The authors have effectively addressed my concerns regarding the evaluation of domain-specific LLMs and sensor-wise analysis, and have provided new insights in their response. I believe these additions further strengthen the contribution of AgroMind, and I am therefore willing to raise my score.

---

> > > ### Author Response · Authors · 2025-08-06
> > >
> > > Thank you very much for your thoughtful and encouraging comments. We truly appreciate your recognition of the revisions we made to improve the clarity and rigor of our benchmark. We are glad that the additional experiments and analyses effectively addressed your concerns regarding domain-specific evaluations and sensor-wise performance breakdowns. We are grateful for your support and recommendation again.

---

### Official Review · Reviewer_UTXy · 2025-07-03

**Rating:** 5
**Confidence:** 4

**Summary:**

The manuscript introduces AgroMind, a comprehensive benchmark for evaluating Large Multimodal Models (LMMs) in agricultural remote sensing (RS). Addressing the scarcity of robust benchmarks, AgroMind integrates eight public and one private dataset, comprising 15,556 images and 25,026 QA pairs across four task dimensions: spatial perception, object understanding, scene understanding, and scene reasoning, covering 13 task types like crop identification and health monitoring. Experiments with 18 open-source and three closed-source LMMs reveal performance gaps in spatial reasoning and fine-grained recognition, with human performance trailing leading models.

**Dataset Code Accessibility:**

Yes

**Dataset Code Comments:**

The dataset is publicly available on Hugging Face, and the code has been released on GitHub with a detailed README. The data and scripts are well-organized and accessible, which facilitates reproducibility and community use. The documentation clearly describes the data structure, usage instructions, and baseline setups.

**Ethical Considerations:**

No, there are no or only very minor ethics concerns

**Final Justification:**

The authors have addressed my issues. I'd like to raise my score.

**Limitations Weaknesses:**

1. Temporal dynamics are crucial for farmland management, yet AgroMind does not address crop phenological changes over time. Additionally, the dataset’s global spatial distribution is unclear, limiting its relevance across diverse agroecological zones. A statistical analysis of global data distribution is recommended.

2. The data pre-processing details mention logical reasoning enhancement, but it is unclear how this is implemented. It is uncertain whether AgroMind includes chain-of-thought (CoT) responses to support reasoning tasks.

3. The evaluation process for open-ended answer types is not specified, leaving it unclear whether models or human judges assess responses and which models are involved, hindering reproducibility in agricultural contexts.

**Strengths Contributions:**

1. AgroMind thoroughly considers diverse answer types (e.g., judgment, counting, open-ended) for benchmark evaluation, incorporating multiple sensor sources (UAV, Satellite, Camera), which enhances its applicability to varied agricultural scenarios like crop monitoring and pest detection.

2. The four task dimensions—spatial perception, object understanding, scene understanding, and scene reasoning—cover critical agricultural tasks such as crop identification, health monitoring, and environmental analysis.

---

> ### Author Rebuttal · Authors · 2025-07-29
>
> We sincerely thank the reviewer for your thoughtful and constructive feedback. We are pleased to know that you recognized the importance and technical rigor of our work. Below, we provide detailed point-by-point responses and clarify all concerns raised.
>
> **Q1: Temporal dynamics / phenological representation; unclear global spatial distribution**
>
> A: Thank you for pointing out these important aspects. We fully acknowledge the importance of representing temporal dynamics and geographical diversity in agricultural benchmarks. AgroMind **indeed incorporates both aspects**, which we will clarify more explicitly in the revised manuscript.
>
> **1. Temporal and phenological information**
>
> AgroMind does **include temporal information**, particularly in tasks such as crop **growth status recognition**, multi-season analysis, and climate impact monitoring. For instance:
> - Growth status recognition: The benchmark contains samples of crops from the seedling stage to the mature stage, with dedicated QA pairs on oil palm growth status, as well as crop-weed competition management during critical phases.
> - Anomaly dynamics prediction: Samples include time-dependent anomalies such as standing water, pest spread, or seasonal weed clusters, enabling models to infer progression patterns.
> - Multi-season imagery: A significant portion of the image data is derived from multi-temporal satellite and UAV imagery, captured across different years and seasons, facilitating implicit phenology modeling.
>
> We will enhance the manuscript by clearly indicating the **temporal labels** embedded in the data and adding examples in Section 3.1 and Appendix B.1.
>
> **2. Global spatial distribution**
>
> To address the spatial coverage concern, we have **conducted the statistical analysis of global data distribution**. AgroMind, intentionally **covers a diverse range of regions**, including China, the United States, the European Union, Brazil, and parts of Africa, to reflect global agricultural diversity in terms of crops, climate, and land use patterns. We acknowledge that some regions (e.g., certain African and Southeast Asian countries) are relatively underrepresented due to limitations in publicly available, high-resolution multimodal agricultural data. However, this does not affect the evaluation of the model performance.
>
> To enhance transparency, we will include **geographical coverage maps** of datasets in Appendix B.1, where the number of samples per region is visually indicated. Furthermore, we explicitly discuss the implications of this imbalance and its potential impact on fairness in Appendix C.5, and we plan to address this limitation by continuously expanding the dataset with better-balanced regional representation in future releases.
>
> **Q2: Logical reasoning enhancement in data pre-processing; chain-of-thought (CoT)**
>
> A: Thank you for this valuable comment. The term "logical reasoning enhancement" refers to our approach of artificially **pre-defining hierarchical reasoning processes** in QA pair design, rather than directly using raw annotations as answers of QA pairs. For instance:
>
> a. Crown diameter calculation (e.g., "What is the crown diameter of the tree marked by the red box?")
> - Pre-defined chain: Locate bounding box → Identify tree → Calculate pixel coverage of crown diameter → Convert to physical scale using metric ratio
>
> b. Weed control decision (e.g., "Propose weeding measures based on crop growth image")
> - Pre-defined chain: Quantify weed coverage → Analyze occlusion rate → Generate actions via agronomic rules (While the original label only has masks for crops and grasses)
>
> As illustrated in the examples above, we adopt predefined reasoning procedures to construct the question-answer pairs, and we expect the models to follow similar reasoning paths when generating responses. This logic enhancement strategy is essentially a preset human reasoning process, where each QA pair is crafted to reflect an implicit multi-step decision process. Such a design encourages models to emulate expert-like, structured reasoning, thereby enabling a more faithful and rigorous evaluation of their logical capabilities. We will clarify this in the Section 3.2.
>
> AgroMind does **not currently provide explicit CoT-style answers** in the QA annotations. The current design focuses on evaluating whether models are capable of producing logically grounded final answers to multi-step reasoning tasks (e.g., tree measurement, weed control decisions), but without enforcing the step-by-step reasoning path in the answer format.
> We agree that explicit CoT supervision can substantially enhance the interpretability and reasoning assessment of multimodal LLMs. As part of our future work, we **plan to enrich AgroMind with CoT-annotated responses** for selected complex reasoning tasks. Specifically:
>
> - We are designing CoT templates that reflect agronomic expert thinking, such as stepwise diagnosis of crop diseases or multi-factor yield analysis.
> - We are collaborating with domain experts to generate manual CoT-style annotations for a subset of the data, which will allow the community to both fine-tune and evaluate LMMs under a more explainable reasoning setting.
>
> **Q3: Evaluation process for open-ended answer types**
>
> A: We sincerely appreciate the reviewer's feedback on the evaluation methodology for open-ended answers. Below, we provide **a detailed clarification of our evaluation process**, including the **evaluation criteria**, the **involved model**, and the **threshold-setting process**, with a focus on ensuring reproducibility in agricultural contexts.
>
> All open-ended questions have clear and fixed reference answers, which are designed or verified by domain experts and serve as the ground truth for **semantic comparison**. Specifically, we use the **Paraphrase-MiniLM-L6-V2** model [1], a lightweight and high-performing sentence embedding model from the Sentence-BERT family, to compute **cosine similarity** between each model-generated answer and the reference answer. Paraphrase-MiniLM-L6-V2 has been extensively trained on large-scale paraphrase datasets and is capable of recognizing semantic equivalence even across diverse linguistic expressions — making it particularly suitable for evaluating semantic consistency in open-ended agricultural QA.
>
> We define an open-ended prediction as correct only when its **semantic similarity score** with the reference answer **exceeds a fixed threshold**. Through **preliminary experiments on a dataset subset**, we determined the **threshold as 0.60**, under which all models achieve reasonable accuracy. We will include details of this evaluation pipeline in **Appendix C.4** of the revised version and provide the open-source scoring script in our released benchmark toolkit for full reproducibility.
>
> **References:**
>
> [1] Reimers N, Gurevych I. Sentence-bert: Sentence embeddings using siamese bert-networks[J]. arXiv preprint arXiv:1908.10084, 2019.
>
> Once again, we sincerely thank *Reviewer UTXy* for the thoughtful and encouraging feedback. We believe your suggestions have significantly improved the clarity and rigor of our work. We are excited about the potential of AgroMind to become a valuable asset for both the LMMs and agricultural remote sensing communities. We will **address all the above issues in the final manuscript** if the paper is accepted. We believe that our revisions and responses demonstrate the strengthened value of this work.

---

> > ### Comment · Reviewer_UTXy · 2025-08-04
> >
> > The authors have addressed my issues.
> > I'd like to raise my score.

---

> > > ### Author Response · Authors · 2025-08-04
> > >
> > > Thank you very much for your positive feedback. Your recognition of our revisions is deeply encouraging, and we appreciate the opportunity to enhance this work under your guidance. We also welcome your participation in further discussions regarding the General Comments.

---

### Comment · Area_Chair_ijMe · 2025-08-08

Dear Reviewers,

During the rebuttal and discussion period, we kindly remind you to carefully read the authors’ responses and actively engage in the discussion when appropriate. This stage is not merely for submitting mandatory acknowledgement or adjusting scores, your thoughtful consideration of the authors’ rebuttals and constructive interaction are also crucial to a fair and thorough review process.

Thank you again for your continued support and contributions.

---

### Note · Authors · 2025-08-12

We sincerely thank the AC and all reviewers for their time, constructive feedback, and thoughtful engagement throughout the rebuttal and discussion process. During this stage, we have carefully revisited every aspect of our work, from **dataset construction** and **evaluation methodology** to **domain-specific model assessment** and **sensor-wise analysis**, ensuring that **each concern raised was thoroughly addressed**. Through clarifications, additional experiments and dataset refinements, we have enhanced both the robustness and the clarity of our contributions. The reviewers have highlighted several strengths of our paper, including:

* The introduction of **AgroMind**, a comprehensive and well-structured benchmark that fills a critical gap in evaluating Large Multimodal Models (LMMs) for the underrepresented domain of agricultural remote sensing (**UTXy**, **bypw**, **yGqA**).
* Multiple reviewers commended the benchmark's rigorous construction, including its **diverse task dimensions**, rich question formats, and inclusion of **multi-sensor data** (satellite, UAV, and camera), which ensures a holistic and realistic assessment of model capabilities (**UTXy**, **bypw**).
* Reviewers noted the detailed experimental evaluation of over 20 models, which offers valuable diagnostic insights into the limitations of current LMMs, especially in fine-grained recognition and complex spatial reasoning (**yGqA**).
* The paper’s commitment to **reproducibility**, through the well-documented public release of the dataset and evaluation code, was recognized as an essential strength of our work (**UTXy**, **yGqA**).

Following our detailed responses and additional results, **all four reviewers confirmed that their concerns were resolved**. Each reviewer raised their scores and expressed strong support for the acceptance of our work, recognizing AgroMind as a comprehensive, multi-scenario benchmark that will serve as a valuable foundation for advancing agricultural remote sensing research with large multimodal models. Reviewers particularly highlighted the hierarchical task design spanning perception to reasoning, and its practical relevance in addressing real-world agricultural challenges.

We greatly appreciate the collaborative and constructive review process, which has strengthened the final version of our paper. We are confident that the revised version is ready for publication and will provide meaningful value to both the agricultural and AI research communities.

---

### Decision · Program_Chairs · 2025-09-18

**Decision:**

Accept (poster)

**Comment:**

This work aims to address the lack of benchmarks for large models in the field of agricultural remote sensing by proposing AgroMind, which involves multiple downstream task labels and contributes positively to advancing the application of large models in this domain. The diversity of the proposed dataset in terms of scenarios and tasks is the main highlight, and the extensive experimental validation is also commendable. On the downside, reviewers primarily raised concerns about the unclear description of dataset construction details and the presence of some duplicated samples. During the rebuttal and discussion stage, all reviewers indicated that the authors had addressed their concerns to varying extents, with some reviewers raising their ratings. Ultimately, all ratings leaned toward the positive. Taken together, I am inclined to recommend acceptance of this work.